# Finding gaps in the national electric vehicle charging station coverage of the United States

Lily Hanig ®[1,2] ✉, Catherine Ledna ®[2], Destenie Nock ®[1,3], Corey D. Harper[3,4], Arthur Yip ®[2], Eric Wood[2] & C. Anna Spurlock ®[5]

The United States federal government has invested $7.5 billion into charging infrastructure, including the National Electric Vehicle Infrastructure Program, to build fast charging stations along designated highways for long-distance car travel. We develop a consecutive coverage metric to compute the percent of United States roads (traffic-weighted) that are consecutively accessible within 500 miles of each county. We answer (1) what the state of consecutive coverage is in each county and (2) what the increase in coverage is when designated highways receive fast chargers. In 2023, 10% of counties had at least 75% minimum viable coverage. We find that if all designated highways receive fast-charging stations, 94% of United States counties will reach at least 75% fast charger coverage. However, the remaining counties are rural. This demonstrates that federal funding for fast chargers will help connect most—but not all—counties to the national network of continuously accessible charging stations.

The global push for decarbonization and technology advancement has led to the rapid expansion of light-duty electric vehicles (EVs) on the market, presenting a lower-emissions option than internal combustion engine vehicles[1]. However, EVs need sufficiently convenient charging infrastructure to approach or surpass the functionality of internal combustion engine vehicles. The literature frequently discusses 'range anxiety'[2,3], which implies concerns over the distance an EV can cover from a single charge. Instead, we use the term 'charging anxiety' to move from a vehicle-centric to an infrastructure-centric approach. Charging anxiety encompasses the sufficiency of EV charging infrastructure to meet changing needs in frequency (gaps in coverage), density (queuing), and reliability (out of service). Although less than 5% of car trips are longer than 30 miles (well within the range of today's EVs) and only 0.1% of all car trips surpass 500 miles[4], charging anxiety for long-distance trips tends to have a disproportionate impact on consumers' vehicle purchases[2,5,6]. Consumers may rarely take long-distance trips but highly value having the option available when choosing a personal vehicle[6–8].

Charging anxiety is a reasonable concern among United States (U.S.) consumers, who are used to the convenience of 110,000 gas stations in the U.S.[9]. Public EV charging stations have become widespread, with 60,000 publicly available or planned Level 2 and DC (direct current) fast charging stations in the U.S.[10]. Level 2 chargers have a 5–19.2 kW power rating (208–240V), the same power used for clothes dryers, and are what is most typically found in dedicated home and workplace chargers. DC fast chargers (also referred to as "fast chargers") have power ratings that can range from 50 kW up to as much as 350 kW[11,12]. However, there is a gap in the research assessing the adequacy of charging station distribution between cities and rural areas. Even when traversing long distances in an EV is possible, it may not be convenient: a driver may need to detour to stations, wait to charge, plan around long gaps between stations, and risk depending on stations that could be out of service. We define minimum viability as a trip with sufficient charging station access to traverse the trip's distance (even if the stations have slow charge speeds). It is worth noting

[1]Engineering and Public Policy, Carnegie Mellon University, 5000 Forbes Ave., Pittsburgh, PA, USA. [2]National Renewable Energy Laboratory, 15013 Denver West Parkway, Golden, CO, USA. [3]Civil and Environmental Engineering, Carnegie Mellon University, 5000 Forbes Ave., Pittsburgh, PA, USA. [4]Heinz School of Policy and Information Systems, Carnegie Mellon University, 5000 Forbes Ave., Pittsburgh, PA, USA. [5]Lawrence Berkeley National Laboratory, 1 Cyclotron Rd., Berkeley, CA, USA. ✉e-mail: lily.r.hanig@gmail.com

that minimum viable charging station coverage for long-distance trips, but with potentially long waits, may be insufficient to spur EV adoption for some consumers[7]. Therefore, our fast charger coverage scenario is the charging scenario most aligned with consumer expectations around refueling for a road trip.

There are several types of EV chargers: Level 1, Level 2, and DC fast chargers. Level 1 typically has a power rating of 1–2 kW (120 V) and is the rating of power used at a typical outlet in a home[11]. Level 2 chargers have a 5–19.2 kW power rating (208–240 V), the same power used for clothes dryers, and are what is most typically found in dedicated home and workplace chargers. DC fast chargers (i.e., fast chargers) have power ratings that can range from 50 kW up to as much as 350 kW[11,12]. Fast chargers are suitable for intercity and long-distance travel when passengers want to arrive at their destination as fast as possible and avoid long waits while their vehicle charges. However, not all charger configurations and vehicles are compatible. For example, Tesla DC fast chargers are not compatible with all EVs, but Tesla has made agreements with other manufacturers to open some Supercharger stations to other vehicles and to share its connector design with other car manufacturers[12,13]. This analysis focuses on nonproprietary Combined Charging System Combo 1 (CCS-1) plugs, the prevalent standard before Tesla made the above changes, and excludes charging stations with proprietary plug types (e.g., Tesla). We run a sensitivity analysis to compare how coverage would improve if proprietary stations were made universally accessible (Supplementary Fig. 1).

The U.S. is investing significantly in charging infrastructure to spur EV adoption. The federal government committed to investing up to $7.5 billion into public EV charging infrastructure through the Bipartisan Infrastructure Law[14,15]. This consists of $5.0 billion for the National Electric Vehicle Infrastructure (NEVI) Formula Program administered by the U.S. Department of Transportation through the states and $2.5 billion for the Charging and Fueling Infrastructure Discretionary Grant Program[14]. Additionally, $3.0 billion in public investment has been made across all levels of government, led by programs from California[14]. The NEVI program first places fast chargers along designated highways referred to as alternative fuel corridors (AFCs), and will place charging stations until all AFCs have sufficient NEVI-compliant stations[14] (defined as four or more DC fast chargers with at least 150 kW at each port within 1 mile of an AFC). AFCs comprise the higher-traffic interstates in the U.S.; however, these are not the only U.S. roads used for long-distance travel. There is a gap between coverage of charging stations along AFCs and the full coverage of charging stations on all U.S. highways. This paper defines an AFC as reaching NEVI compliance if charging stations are spaced every 50 miles or less with at least four universal-plug DC fast chargers[14]. Coordination across states will help achieve nationwide coverage. Some states, such as California, have high coverage already; however, good coverage in neighboring states and along highways connecting multiple states is needed to drive seamlessly across the country.

The number of charging stations is not as meaningful as the consecutive coverage from a starting point because if there is too large of a gap between stations, people still cannot drive long distances in an EV. We define EV traversability as having a fast charger on every consecutive 50-mile segment, which aligns with the NEVI program definition. We create a consecutive coverage metric, which measures the percent of National Highway System (NHS) roads (traffic-weighted) that are consecutively accessible within 500 miles of each starting county. Our metric excludes roads that cannot be reached due to a large gap (50 miles or more) in charging station access (e.g., just because a road has a charging station does not mean it can be reached from a given location before hitting a long stretch without a charging station). Our metric also compares coverage for long-distance trips between different counties. We use our consecutive coverage metric to answer several questions: (1) What is the current state of consecutive EV charging coverage from each county in the U.S.? (2) What is the

increase in coverage per state when all AFCs are built out with fast charging stations? (3) Does the Bipartisan Infrastructure Law funding distribution improve access evenly across states (allowing each state to reach the same level of coverage)? Charging anxiety (sometimes referred to as 'range anxiety') is one of the primary concerns among consumers[2,7,8,16,17]. Charging anxiety decreases as drivers become accustomed to driving EVs[18]; however, coverage is also an important metric for reliability. Without sufficient charging infrastructure coverage, an out-of-service charging station can stop even an experienced EV driver from completing a long-distance trip. Reliability perceptions are essential not just for EV adoption, but to maximize EV utilization[5,18].

Several papers have measured perceptions of charging anxiety and how it inhibits EV purchasing. Franke and Krems (2013) define charging anxiety as the minimum state of charge a consumer is willing to reach before becoming prohibitively anxious[18,19]. They also found that at 15% state of charge, more non-EV drivers choose to charge than not. Other papers assess the ideal distance between charging stations[2]. Pevec et al. (2020) found that the mean ideal distance between charging stations (in towns and cities) for EV owners and non-EV owners is 7 km (4.3 miles). Their survey results show that the ideal distance between gas stations and charging stations are similar[2], while Melaina et al. (2013) surveyed the density of chargers consumers desire in a 100-square-mile area[7]. We build from these studies by using the NEVI program mandate that requires a charging station at least every 50 miles to evaluate consecutive coverage.

Increased frequency of charging stations can help alleviate charging anxiety and increase the utility of EVs[16,20,21]. We quantify the sufficiency of infrastructure development to maximize the benefits of EV adoption, which several papers have highlighted. Egbue and Long (2012) found that a perceived lack of public charging infrastructure is one of the top barriers to EV adoption[16]. Sierzchula et al. (2014) found that public charging infrastructure was the strongest indicator of EV adoption[20]. Similarly, Higueras-Castillo et al. (2021) found that range and reliability are top predictors for EV purchasing among potential consumers surveyed in Spain[22]. For non-passenger vehicles, Konstantinou and Gkritza (2023) surveyed truck drivers to investigate motivations for electric truck adoption, finding that charging time, product availability, and financial viability were among the largest concerns, as opposed to charging access[23]. Several papers also assess the value of community-level charging stations[3,24]. Almeida Neves et al. (2019) used regression to study the factors influencing EV adoption in 24 EU countries, finding that access to charging stations drives adoption across types of EVs[25]. Comparatively, Vergis and Chen (2015) found access to charging stations to be a leading predictor of battery-EV adoption but a less significant factor for plug-in hybrid EV adoption[26].

Charging station development and EV adoption require coevolution, as discussed by Muratori et al. (2020)[5]: Insufficient utility of a charging station (due to insufficient charging demand) leads to high electricity costs at charging stations[5,27]. However, if EV adoption outpaces public charging station development, EV users may experience charging anxiety and lower overall utility[5,18]. Lanz et al. (2022) find that the levelized cost of electricity of public chargers in Europe decreases as charging station utilization rates increase, further demonstrating the need for EV infrastructure coevolution[28]. Neubauer and Wood (2014) found that adding infrastructure increases EV utility, but lower-cost options exist to decrease charging anxiety: improving travel and battery range predictions[18]. This aligns with Rauh et al.'s (2015) findings that experienced EV drivers are willing to drive farther on a single charge, as they better predict range[21].

Several papers use optimization to place charging stations. Bräun et al. (2020) and Jochem et al. (2019) optimized placement for long-distance travel in Australia and Europe[29,30]. Xie et al. (2018) used a genetic algorithm to optimize the placement of intercity chargers in California based on trip origin-destination pairs[31]. Xu et al. (2020)

optimized placement given demand flow between intercity origin-destination pairs[32]. These papers optimize the number of charging stations needed in a network or between origin-destination pairs, but they do not compare station access across different regions, nor do they compute improved access. Instead, we assume consumers need a certain charging station frequency to feel comfortable completing a trip. Our work evaluates the sufficiency of planned or existing charging station infrastructure rather than optimizing infrastructure placement.

Several papers assess the value consumers place on the ability to take trips beyond the range of a typical EV. Melaina et al. (2013) conducted a discrete choice survey and found diminishing cost penalties for the inability to complete long-distance trips in an EV[7]; this study demonstrates that although only 0.1% of trips are farther than 500 miles[4], having the option to travel long distances matters to consumers. The inability to do so imposes significant ($1000–$2000) cost penalties. Hidrue et al. (2011) looked at the value of extending EV battery range to consumers, finding that survey respondents were willing to pay up to $75 per additional mile of driving range, with a decreased willingness to pay as the range increases[33]. We define the long-distance range we consider as 500 miles given the importance long-distance viability is to consumers[7], combined with the infrequency of consumers traversing the entire U.S.

Decarbonizing the light-duty vehicle sector is imperative to meet climate goals[34,35]. Hoehne et al. (2023) outlined pathways to decarbonize U.S. passenger and freight vehicles using the National Renewable Energy Laboratory's (NREL's) Transportation Energy & Mobility Pathway Options (TEMPO) model for EV adoption and associated emissions under varying assumptions. Charging station infrastructure is an important driver of EV adoption[1], but is just one tool available for decarbonizing transportation[36–38]. Mulrow and Grubert (2023) highlighted the potential impact of changing behavior, such as decreasing total miles traveled, on overall vehicle emissions[36]. Aguilar et al. (2024) found that if 50% of EVs in Europe implemented vehicle-to-grid technology, they could meet the full demand for battery storage in Europe, reducing infrastructure build-out[39]. Ren et al. (2023) found that although EVs have fewer emissions than internal combustion engine vehicles over the vehicle lifetime, their greenhouse gas emissions are front-loaded due to battery manufacturing[40]. Therefore, efforts should strategically replace high-emitting and high-mileage internal combustion engine vehicles with EVs[40].

The literature quantifies charging station coverage metrics by measuring the percentage of inaccessible trips given station access or by quantifying the number of stations lacking[3,41]. Melliger et al. (2018) show that in Finland and Switzerland, more public charging infrastructure near homes would allow 99% of trips to be completed in an EV[3]. They define coverage by the percentage of annual trips that can be completed in an EV. Most U.S. passenger vehicle trips are local, and therefore can be completed in an EV; however, the ability to complete a long-distance trip has an outsized impact on a consumer's choice to purchase a vehicle[7]. A study by NREL found that a few hundred fast chargers are required to provide minimum coverage between cities; however, for more rural communities, about 8000 charging stations would be needed[41]. This computes the chargers needed to reach full coverage but does not compute the overall coverage for a given urban area. Instead, we look at coverage based on the ability to complete trips of 500 miles from any given starting county in the U.S.. We differentiate our metric by computing consecutive coverage rather than the overall percentage of roads with a charging station or percent of traversable trips annually.

NREL has several tools for assessing EV charging station needs in communities and for long-distance trips. The Electric Vehicle Infrastructure - Projection (EVI-Pro) tool can be used to find the total charging stations needed for a metropolitan statistical area and the associated electricity demand[42]. NREL's EVI-X modeling tools help address EV charging station questions from different angles, such as charging station need at the community level (EVI-Pro), siting charging stations for long-distance trips (EVI-RoadTrip), and planning charging station design[43]. Wood et al. (2023) used the EVI-X suite of models to find the build-out of charging station equipment needed to meet forecasted EV adoption out to 2030[44]. EVI-Pro estimates the number of charging stations needed in a metropolitan area to meet community-level charging demand, while our metric assesses the ability to drive long distances without hitting gaps in coverage. Additionally, our metric is at the county level for all U.S. counties, while EVI-Pro Lite is limited to metropolitan statistical areas. EVI-RoadTrip is a useful model for assessing charging station adequacy for a specific route[43]. In contrast, our metric shows regional charging station adequacy and is useful for comparing across counties, states, and policies.

The TEMPO model is an energy systems model of the U.S. transportation system[45]. Among other features, TEMPO estimates vehicle stock, new technology adoption (including types of EVs), activity, and energy consumption for the entire U.S. light-duty vehicle fleet[45]. TEMPO is an important model for assessing the potential for charging station coverage, such as our consecutive coverage metric, to find the change in vehicle adoption and demand due to infrastructure improvements[45,46]. Hoehne et al. (2023) used TEMPO to project passenger and freight decarbonization and model the emissions outcomes under varying scenarios such as tightened fuel standards, zero-emissions vehicle mandates, and lowering miles traveled[46]. The model considers charging station access when modeling EV adoption propensity and associated changes in charging load. Therefore, our metric could be used with a model such as TEMPO to inform their adoption propensity with a more nuanced assessment of long-distance charging station coverage.

In this work, we develop a consecutive coverage metric for long-distance charging stations at the county level to show the current state of consecutive charging station access around the U.S. and the anticipated state of coverage if fast charging stations are placed along AFCs. For light-duty vehicles, we find that the current state of fast charging station access is low; however, once all AFCs reach NEVI compliance, 94% of U.S. counties will reach consecutive charging station coverage at 75% or higher.

## Results

We present results on consecutive coverage for three scenarios at the state and county levels: any charging station with Level 2 or DC fast chargers, Minimum Viable Coverage; the state of coverage only considering charging stations with at least four DC fast chargers, Fast Charger Coverage; and the scenario when all AFCs have DC fast chargers that are NEVI-compliant, AFCs Reach NEVI-Compliant Status, as well as the increase in coverage from all AFCs reaching NEVI compliance. We consider coverage up to a radius of 500 miles from the origin for all scenarios. Additionally, we vary the distance in a sensitivity analysis (see Supplementary Fig. 2).

### Consecutive coverage metric

We develop a metric to quantify consecutive national EV charging station coverage for long-distance trips. We use a breadth-first search function: an algorithm that searches along a tree data structure (e.g., a network of road segments) for a charging station and continues to search along all connected segments that contain a charging station. This compares long-distance traversability for EVs from every county in the U.S. (outside of a 50-mile buffer of each county population center). In the breadth-first search, we use geospatial highway data coupled with public EV charging station location data to find the percent of highways (traffic-weighted) that are consecutively accessible to public charging stations from a given starting county without gaps of

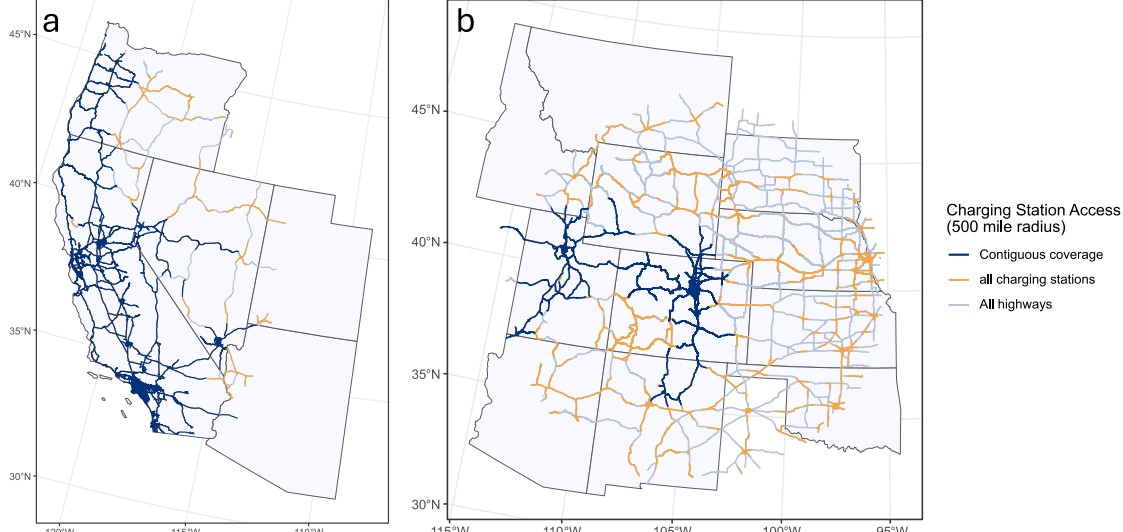

**Fig. 1 | Consecutively accessible highway segments. a** Map of consecutively accessible highway segments from San Francisco, California (97% coverage) while considering planned and publicly available Level 2 and direct current fast (universal plug) charging stations within 1 mile of each segment. **b** Map of consecutively accessible highway segments from Denver, Colorado (39% coverage), while considering planned and publicly available Level 2 and direct current fast (universal plug) charging stations within 1 mile of each segment. Charging station data were retrieved on March 1, 2023. Several sections of highways contain charging stations along them; however, there is a gap too large between charging stations for the section of highway to be accessible from the starting county.

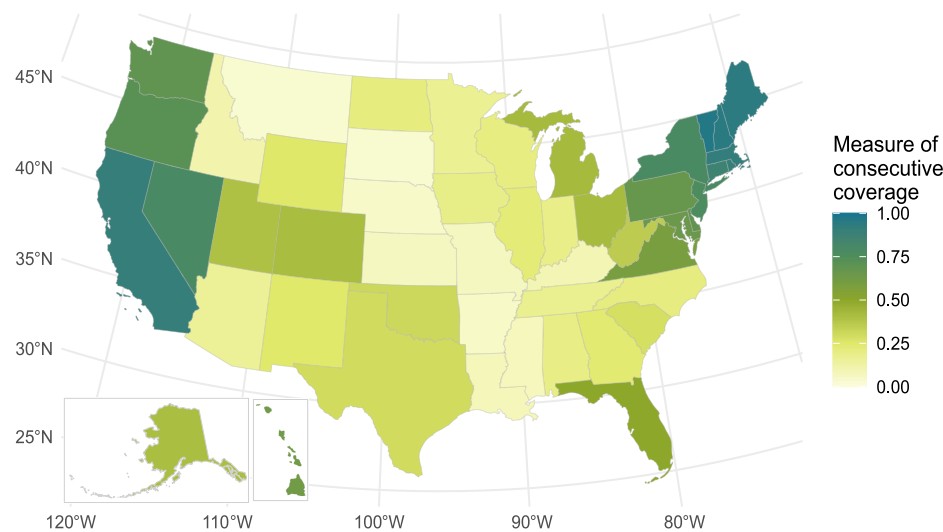

**Fig. 2 | State-level minimum viable coverage.** Map of current (2023) state-level Minimum Viable Coverage weighted by vehicles registered per county. California, Nevada, and the Northeast reach high coverage (above 75%) Source data are provided as a Source Data file.

more than 50 miles. Road segments are weighted by their annual average daily traffic (daily traffic volume averaged over a full year)[47]. Additionally, we exclude truck-specific traffic for the consecutive coverage metric when calculating light-duty vehicle traffic.

We demonstrate the consecutive coverage metric with two sample counties showing the specific routes that are consecutively accessible. In Fig. 1, we show the map of consecutively accessible roads (with charging stations) from San Francisco, California (Fig. 1a), and from Denver, Colorado (Fig. 1b). Both maps show NHS road segments that are consecutively accessible from the given starting counties (dark blue), have a charging station but are not accessible (orange), and do not have a station (gray). Figure 1a shows that San Francisco has high consecutive coverage across the state (97%), but less access to the more rural parts of Nevada and Oregon. Despite this, the county still

reaches 97% consecutive coverage because rural roads tend to have lower traffic.

### State minimum viable coverage (2023)

Figure 2 shows the current (2023) state of coverage in the U.S. when considering both Level 2 and DC fast charging stations. This scenario represents travel viability; however, there may be long wait times. In Fig. 2, California, Nevada, and all states in New England have average consecutive coverage above 80%, and states from Louisiana north through Montana have consecutive coverage below 25%. Comparatively, in Fig. 3, when only DC fast charging stations are considered, coverage hovers at or below 30% for all states except California (79%), Nevada (71%), Rhode Island (39%), and Washington (49%).

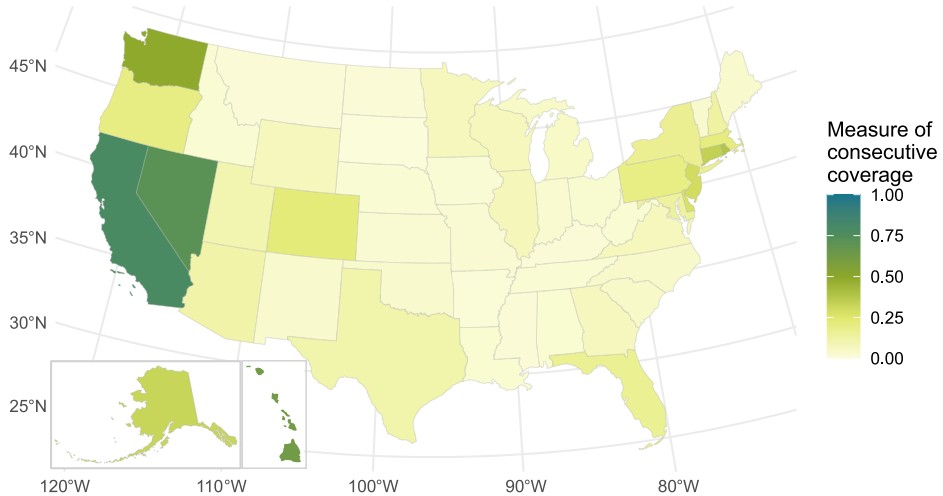

**Fig. 3 | State-level fast charger coverage.** Map of current (2023) fast charger coverage at the state level weighted by vehicles registered per county. Only California and Nevada reach fast charger coverage above 50%. Source data are provided as a Source Data file.

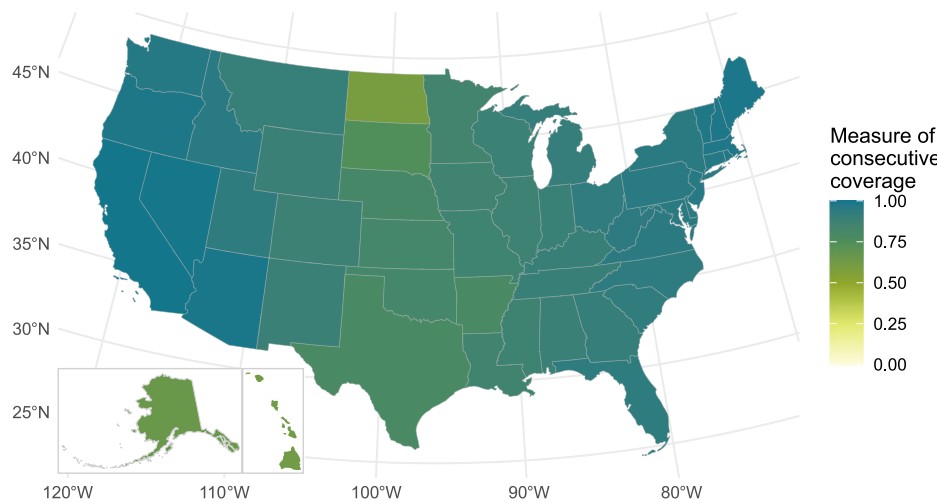

**Fig. 4 | State-level fast charger coverage of alternative fuel corridors (AFC).** Fast charger coverage when all AFCs reach National Electric Vehicle Infrastructure Program (NEVI) compliance. All states reach coverage of 60% or higher when AFCs reach NEVI-compliant status. Source data are provided as a Source Data file.

## State fast charger coverage (2023)

Figure 3 shows the state-level coverage of fast chargers in the Fast Charger Coverage scenario. New England only reaches 39% coverage (in Rhode Island) when considering DC fast chargers. This demonstrates that New England, despite having good minimum viable coverage ( >60%), lacks infrastructure that can handle high demand or avoid longer wait times.

## State AFCs reach NEVI-compliant status

Figure 4 shows the state-level coverage of fast chargers when all AFCs reach NEVI-compliant status. North Dakota reaches the lowest coverage with a vehicle-weighted average coverage of 60%, followed by South Dakota (74%), Arkansas (79%), and Texas (79%). The Northeast, California, Nevada, and Arizona reach above 95%. Coverage increases between 20% and 94% across all states, with California gaining the least because its 2023 fast charger coverage is already 79%.

Table 1 shows the state-level coverage in the U.S. under each scenario and the percent of current (2023) coverage gained from surrounding states. For minimum viable coverage in 2023, Vermont has higher coverage than California (96% and 90%, respectively), but 99% of Vermont's coverage is from neighboring states, compared to 1% for California. The states with the highest out-of-state spillover are

generally smaller; however, some geographically large states still benefit significantly from their neighbors, including Nevada (90% from neighboring states), Pennsylvania (85%), and Oregon (76%). We define out-of-state spillover as the percentage of coverage from charging stations on roads in another state but within 500 miles of a given county and consecutively covered.

Table 1 also shows NEVI-compliant Fast Charger Coverage in 2023. South Dakota has the lowest fast charger coverage at 1%. States with low fast charger coverage have few charging stations and low out-of-state spillover. Florida has the lowest out-of-state spillover (excluding Alaska and Hawaii), with only 8% of coverage from surrounding states, followed by California and Texas at 19%. Alaska and Hawaii have zero spillover due to no road connection to other states within 500 miles. Hawaii is an outlier of low coverage (37%) due to inaccessible charging stations on neighboring islands; for this reason, we exclude Hawaii from Table 1.

For Minimum Viable Coverage, 18 states have at least 50% coverage; however, when only considering NEVI-compliant charging stations, only California (79%) and Nevada (71%) have coverage above 50%. In the AFCs Reach NEVI-Compliant Status scenario, all states reach coverage above 50%, with California and Nevada reaching 99% coverage and North Dakota reaching the lowest coverage (60%). The

**Table 1 | Consecutive coverage by state**

| State | Minimum Viable Coverage | Out-of-State Spillover | Fast Charger Coverage | AFCs Reach NEVI-Compliant Status | Additional Fast Charger Coverage from AFCs |
|---|---|---|---|---|---|
| Alabama | 0.20 | 0.16 | 0.03 | 0.89 | 0.86 |
| Alaska | 0.41 | 0.00 | 0.35 | 0.65 | 0.30 |
| Arizona | 0.16 | 0.05 | 0.12 | 0.98 | 0.86 |
| Arkansas | 0.05 | 0.03 | 0.02 | 0.79 | 0.78 |
| California | 0.90 | 0.11 | 0.79 | 0.99 | 0.20 |
| Colorado | 0.42 | 0.16 | 0.22 | 0.88 | 0.65 |
| Connecticut | 0.87 | 0.82 | 0.34 | 0.96 | 0.62 |
| Delaware | 0.68 | 0.67 | 0.28 | 0.95 | 0.68 |
| District of Columbia | 0.68 | 0.68 | 0.11 | 0.95 | 0.84 |
| Florida | 0.50 | 0.04 | 0.17 | 0.93 | 0.76 |
| Georgia | 0.24 | 0.15 | 0.07 | 0.90 | 0.83 |
| Idaho | 0.11 | 0.09 | 0.03 | 0.95 | 0.92 |
| Illinois | 0.22 | 0.14 | 0.08 | 0.87 | 0.79 |
| Indiana | 0.19 | 0.17 | 0.03 | 0.89 | 0.86 |
| Iowa | 0.19 | 0.16 | 0.02 | 0.85 | 0.83 |
| Kansas | 0.07 | 0.04 | 0.03 | 0.82 | 0.79 |
| Kentucky | 0.10 | 0.07 | 0.02 | 0.90 | 0.88 |
| Louisiana | 0.08 | 0.05 | 0.03 | 0.85 | 0.82 |
| Maine | 0.94 | 0.93 | 0.04 | 0.98 | 0.94 |
| Maryland | 0.65 | 0.58 | 0.14 | 0.95 | 0.80 |
| Massachusetts | 0.92 | 0.81 | 0.21 | 0.97 | 0.76 |
| Michigan | 0.42 | 0.33 | 0.05 | 0.91 | 0.85 |
| Minnesota | 0.17 | 0.03 | 0.08 | 0.84 | 0.77 |
| Mississippi | 0.08 | 0.06 | 0.02 | 0.86 | 0.85 |
| Missouri | 0.07 | 0.03 | 0.03 | 0.86 | 0.83 |
| Montana | 0.03 | 0.01 | 0.02 | 0.90 | 0.88 |
| Nebraska | 0.05 | 0.02 | 0.03 | 0.82 | 0.79 |
| Nevada | 0.79 | 0.72 | 0.71 | 0.99 | 0.28 |
| New Hampshire | 0.95 | 0.92 | 0.14 | 0.98 | 0.84 |
| New Jersey | 0.77 | 0.66 | 0.29 | 0.96 | 0.66 |
| New Mexico | 0.25 | 0.22 | 0.04 | 0.89 | 0.85 |
| New York | 0.78 | 0.64 | 0.17 | 0.95 | 0.78 |
| North Carolina | 0.21 | 0.12 | 0.04 | 0.93 | 0.89 |
| North Dakota | 0.21 | 0.18 | 0.02 | 0.60 | 0.58 |
| Ohio | 0.42 | 0.35 | 0.04 | 0.92 | 0.89 |
| Oklahoma | 0.31 | 0.26 | 0.04 | 0.82 | 0.78 |
| Oregon | 0.71 | 0.55 | 0.20 | 0.98 | 0.77 |
| Pennsylvania | 0.66 | 0.57 | 0.20 | 0.95 | 0.75 |
| Rhode Island | 0.90 | 0.88 | 0.39 | 0.97 | 0.59 |
| South Carolina | 0.29 | 0.25 | 0.05 | 0.92 | 0.87 |
| South Dakota | 0.03 | 0.02 | 0.01 | 0.74 | 0.73 |
| Tennessee | 0.18 | 0.14 | 0.03 | 0.90 | 0.87 |
| Texas | 0.30 | 0.05 | 0.11 | 0.79 | 0.68 |
| Utah | 0.40 | 0.28 | 0.10 | 0.93 | 0.83 |
| Vermont | 0.96 | 0.96 | 0.04 | 0.97 | 0.93 |
| Virginia | 0.59 | 0.52 | 0.08 | 0.94 | 0.87 |
| Washington | 0.69 | 0.26 | 0.49 | 0.96 | 0.47 |
| West Virginia | 0.36 | 0.35 | 0.03 | 0.94 | 0.91 |
| Wisconsin | 0.21 | 0.15 | 0.08 | 0.88 | 0.80 |
| Wyoming | 0.25 | 0.24 | 0.08 | 0.88 | 0.81 |

The out-of-state spillover is the percentage of the 2023 Minimum Viable Coverage from out-of-state investments. Alternative Fuel Corridors (AFCs) reach National Electric Vehicle Infrastructure (NEVI) Compliance AFCs with full coverage of NEVI-compliant stations.

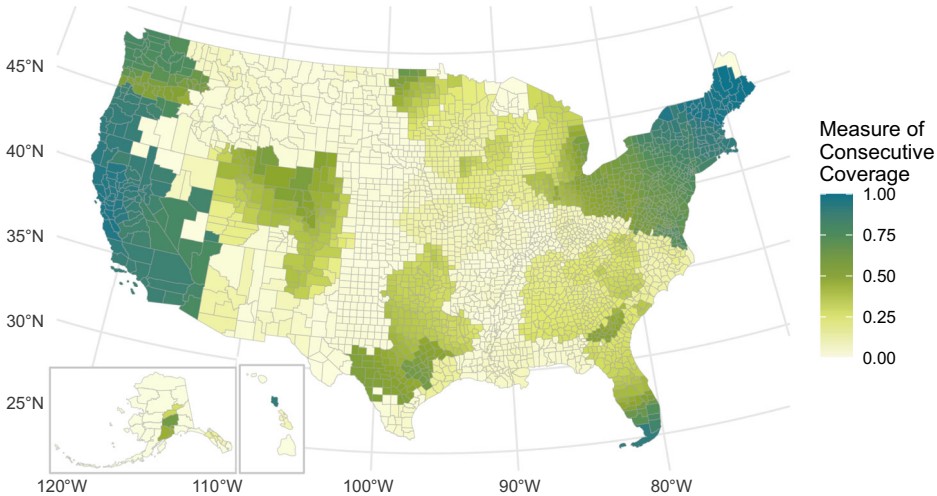

**Fig. 5 | County-level minimum viable coverage.** Charging station consecutive coverage for all Level 2 and direct current fast chargers within 1 mile of a National Highway System road, Minimum Viable Coverage scenario. The West Coast and Northeast reach coverage above 75%, several metropolitan areas reach coverage around 50%, and rural parts of the country remain largely below 25% coverage.

states that gain the least already have great coverage, like California. Vermont and Idaho gain the most coverage from the NEVI program due to the high rate of AFC-designated roads.

### County minimum viable coverage (2023)

The Minimum Viable Coverage metric measures the consecutive coverage from each county in the U.S., including Level 2 and DC fast charging stations in 2023. Figure 5 represents the percentage of roads accessible within 500 miles of each county as a starting location in an EV (consecutive coverage). A county can build out a high saturation of chargers inside its borders; however, if surrounding states do not build out infrastructure, the overall coverage will still be low because local vehicles will lack farther away charging station access.

Figure 5 shows that the West Coast, New England, and South Florida have high minimum viable coverage (greater than 75%). Most major cities reach coverage of at least 30%–50%. Rural counties have significantly lower coverage (typically below 20%). In the Western U.S., the highway network is less dense; therefore, fewer charging stations are needed, leading to counties in Minnesota, Oregon, Washington, New Mexico having coverage above 25% without having major urban areas.

### County fast charger coverage (2023)

Figure 6 shows consecutive coverage from each U.S. county for fast chargers (at least four nonproprietary DC fast chargers) in 2023. This only includes charging stations that would provide a closer experience to refueling internal combustion engine vehicles (e.g., minimal queuing for chargers). The U.S. has fast charger consecutive coverage below 25% with exceptions for California, Nevada, Washington, and some moderate (around 30%) coverage around New York City and Boston. This could be a barrier to EV adoption if drivers highly value fast refueling for long-distance travel.

### County AFCs reach NEVI-compliant status

We model a scenario when all designated AFCs have charging stations with at least four nonproprietary plug configuration DC fast chargers within 1 mile of the highway at least every 50 miles. The Bipartisan Infrastructure Law has allocated $5 billion to place NEVI-compliant stations along AFCs. The program is designed to take 5 years beginning in 2023. However, fully building fast chargers along all AFCs in rural areas with limited grid access may take longer[14]. We model all planned and existing charging stations in 2023 and add coverage along all AFCs

to understand if areas lack coverage when all AFCs reach NEVI-compliant status. We also assess how much each state improves in coverage. We do not model additional charging stations from the NEVI program within communities; instead, we analyze increased coverage along AFCs because our metric is for long-distance coverage.

Figure 7 shows the map of consecutively accessible road segments within 500 miles for each U.S. county in this scenario. To reach this status, an additional 1900 chargers are needed throughout the U.S. Figure 7 shows that when AFCs reach NEVI compliance, coverage of DC fast chargers increases substantially across the U.S. (97% of counties have fast charger consecutive coverage above 50%, compared to 5% of counties in 2023), including in rural areas. However, some rural counties in the Great Plains (from Texas through North Dakota) continue to have fast charger consecutive coverage below 20%. These county centers are more than 50 miles from the nearest AFC or fast charger. This demonstrates that AFCs will help connect most—but not all—rural counties to the national network of continuously accessible charging stations.

## Discussion

We develop a consecutive coverage metric for EV charging stations over long-distance trips (up to 500 miles). We apply our metric to all U.S. counties and aggregate it by vehicle registrations at the state level. To reach NEVI compliance, 1,900 road segments need NEVI-compliant stations along AFCs, bringing 94% of U.S. counties to consecutive coverage above 75% (considering all roads within 500 miles). For all counties to reach 100% fast charger coverage beyond AFCs, 4500 additional road segments need NEVI-compliant charging stations (four universal-plug DC fast chargers), primarily in rural counties. This gap between full coverage of AFCs and full coverage on NHS roads represents 24% of NHS traffic. This requires 4500 additional charging stations because extending coverage into rural areas provides coverage on low-traffic roads. AFCs comprise interstates and other high-traffic roads, which are low-hanging fruit for reaching consecutive charging station coverage.

If the Tesla charging network becomes universally accessible (by adding Magic Docks), it could result in 500 fewer fast charging stations needed (compared to 1900 needed in 2023) to provide full consecutive coverage along AFCs, saving $166 to $332 million in NEVI program funding (Supplementary Fig. 1). Overall, in 2023, fast charger coverage is much higher along the East and West Coasts. Future work could include a cost-benefit analysis of investments in charging stations due to diminishing coverage as interstates reach full coverage.

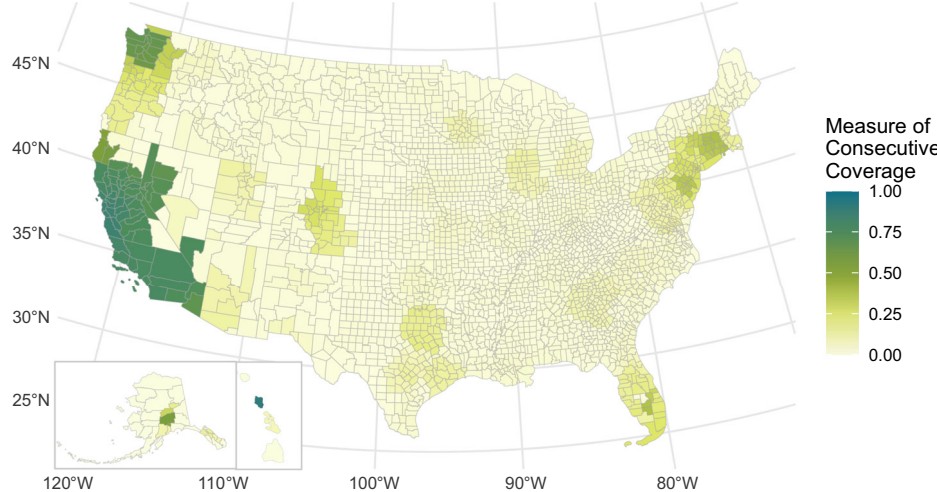

**Fig. 6 | County-level fast charger coverage.** Charging station coverage for charging stations with at least four universal-plug direct current fast chargers within 1 mile of a National Highway System road. Only California, Nevada, and Washington counties reach coverage above 75%.

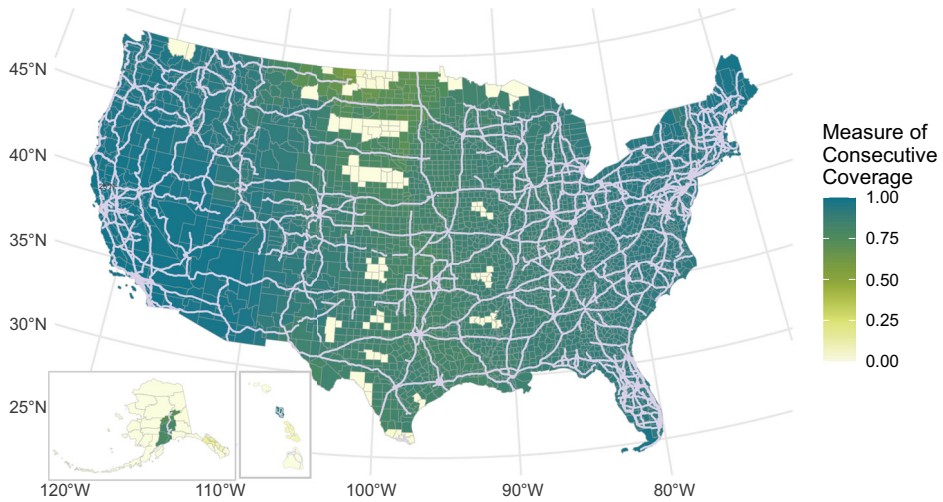

**Fig. 7 | County-level fast charger coverage for alternative fuel corrdiros (AFC).** County-level Fast Charger coverage when all AFCs reach National Electric Vehicle Infrastructure program (NEVI) compliance. AFCs are shown in light purple to highlight gaps in coverage in some rural areas. Pockets of rural counties inaccessible from the AFCs are left with very low coverage.

We caveat that our estimates of charging station coverage do not account for reliability and may exclude more recent investments. Additionally, models of EV adoption, such as NREL's TEMPO model, could be used to extend our consecutive coverage metric to find the influence on EV adoption of charging station coverage and the resulting change in lifecycle emissions from higher EV adoption, including air pollutants.

Charging station access for freight lags behind that of light-duty vehicles, despite heavy-duty trucks compromising a significant portion of interstate traffic[47]. California reaches the highest state-level charging station coverage for both medium- and heavy-duty coverage (24%), while 40 states have heavy-duty coverage below 10% (Supplementary Fig. 3-5). Fast charging coverage increases with median household income at the county level (Supplementary Fig. 6). Future work should investigate the equity implications of long-distance charging station access on EV adoption accessibility (see Supplementary Discussion).

This analysis can inform policy development in several ways. The impact of AFCs can be assessed by considering how reaching NEVI-compliant status will improve consecutive coverage at the state level. The coverage metric allows for easier comparison of impact between states. Building out EV infrastructure requires collaboration on multiple levels of government, mainly because the choices of one state can affect the coverage level of its neighbors. Access to the same information as a starting point for decision-making can help facilitate action that considers impacts beyond a state's borders. Comparing how places prioritize placements can also improve policymakers' awareness of the long-term implications and optimize timeline development. Our coverage metric can provide a dynamic understanding of which states may need more roads to be designated as AFCs and the degree to which states benefit from their neighbors designating new roads as AFCs. We recommend devoting further resources to identifying rural areas that have less infrastructure from the start.

## Methods
Our analysis quantifies consecutive national EV charging station coverage for long-distance trips. We use geospatial highway data coupled with public EV charging station location data in a breadth-first search function (an algorithm that searches along a tree data structure—in this case, a network of road segments—for a charging station and continues to search along all connected road segments of each road segment that

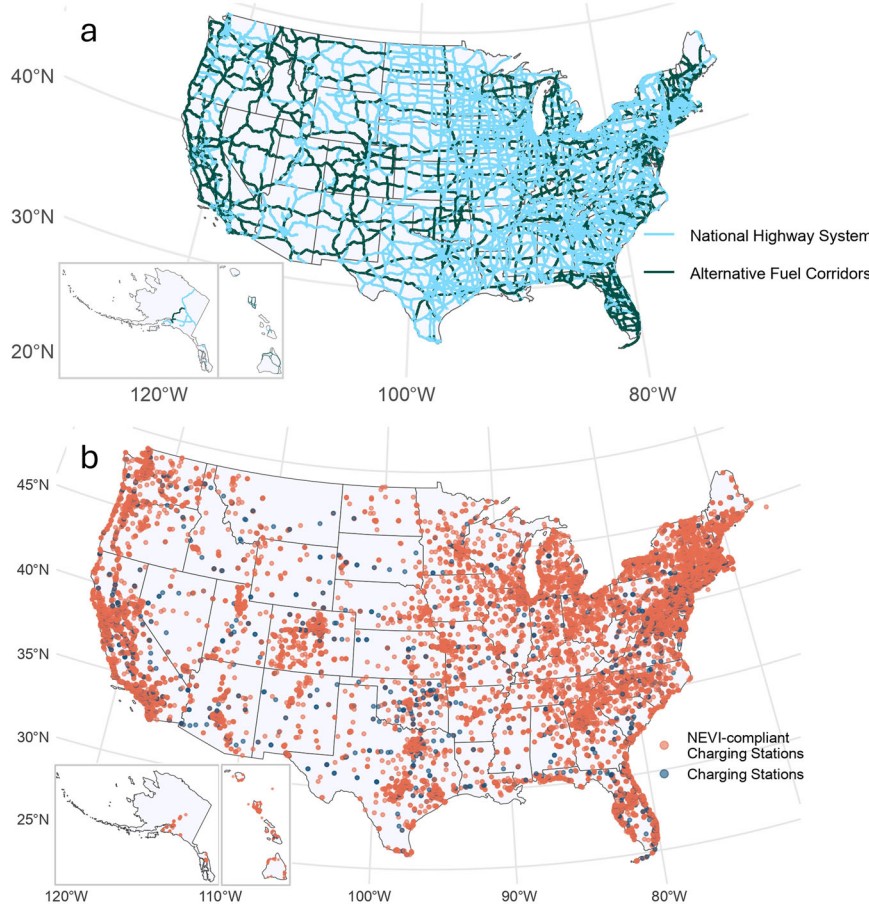

**Fig. 8 | Public charging stations & highways. a** All Alternative Fuel Corridor (AFC) designated highways are in dark green and all National Highway System (NHS) highways are in light blue. **b** Public electric vehicle charging stations in the U.S. as of March 1 2023. Light red shows all stations with at least Level 2 or Direct Current (DC) fast chargers (Level 1 chargers are excluded), and dark blue dots show all National Electric Vehicle Infrastructure program compliant (i.e., four non-proprietary DC fast chargers).

contains a charging station) for each county to calculate a consecutive coverage metric, which provides an estimate of long-distance traversability for EVs. We do this to find the percent of highways (weighted by traffic) that are consecutively accessible to public charging stations from a given starting county without gaps of more than 50 miles.

The Federal Highway Administration has created the NEVI Formula Program, which designates certain highways as a priority for receiving stations (called AFCs). We use AFC designations as of July 6, 2022[48], to model future coverage in a scenario with completed Bipartisan Infrastructure Law investments on AFCs. The map of the AFCs does not perfectly align by name or geospatial data with the NHS dataset. Therefore, we intersect the NHS road segments with a 1-mile buffer of the AFC road segments and match by name to identify which NHS roads are designated as AFCs. Figure 8a shows the map of NHS roads (blue) and AFC-designated NHS roads (dark green). We include the 31 AFC non-NHS roads in the U.S. in this analysis and assign the average traffic of all connected road segments to each non-NHS AFC.

**Data summary**

The data sources used in this analysis include NHS roads, AFC designations, the locations of existing public charging stations, and county-level census data[49]. We use national data sources for the consistency of metrics across states. The NHS data contain the annual average daily traffic for each road segment. These small road sections (often less than a mile) are grouped into 27,000 roads and split into segments every 50 miles. The AFCs used in this analysis are designated as electric for Round 6 of AFC designations[48]. U.S. Department of Energy public

EV charging stations (existing and planned) provide the location, charger type (e.g., Level 2, DC fast charger), provider, and number of chargers per station. Starting at each county's population center, we use a 50-mile radius to find all possible starting segments for a trip from a given county. We conduct this analysis at the county level, including NHS segments within 500 miles of each county's mean population center for all U.S. counties. The 500-mile range stems from less than 0.1% of all passenger trips being farther than 500 miles[4].

The NEVI program requires AFCs to have a charging station every 50 miles to be considered fully built out. We define NEVI-compliant stations as any station with four or more nonproprietary DC fast chargers. The NEVI program also requires each charger to have a minimum power of 150 kW and for stations to be accessible per the Americans with Disabilities Act; however, the U.S. Department of Energy charging station data do not include the power rating or accessibility status, and thus these criteria are excluded from the analysis.

In our analysis, roads that are less than 10 miles in length get flagged as too short to require a charging station (e.g., not every 7-mile road needs a charging station), but are included in the dataset for the continuity constraint so that two highways connected by a short road are still accessible to each other in the breadth-first search function. If the last road segment along a longer road is less than 10 miles (e.g., a 208-mile road split into four 50-mile segments and one 8-mile segment), the short, remainder segment gets appended to the previous segment to avoid requiring additional chargers for the last snippet of an otherwise long highway. There are several types of EV chargers: Level 1, Level 2, and DC fast chargers. Level 1 typically has a power

rating of 1–2 kW (120 V) and is the rating of power used at a typical outlet in a home[11]. Level 2 chargers have a 5–19.2 kW power rating (208–240 V), the same power used for clothes dryers, and are what is most typically found in dedicated home and workplace chargers. DC fast chargers (i.e., fast chargers) have power ratings that can range from 50 kW up to as much as 350 kW[11,12].

The 27,000 grouped road segments in the processed NHS data are labeled as containing a charging station if a public or planned EV charging station[50] is within 1 mile of the road segment. Figure 8b shows the map of charging stations in the U.S. Department of Energy data source[50] built out and operational or planned as of March 1, 2023. All public nonproprietary charging stations that are Level 2 or higher power are mapped in light red, and all NEVI-compliant stations with at least four DC fast chargers (41%) are shown in dark blue. For more details on the data processing methodology see Supplementary Fig. 7.

### Consecutive coverage metric formulation

For each scenario and county, we create a graph data structure out of all eligible segments, and then run a breadth-first search function outward using all road segments within 50 miles of the county population center as starting seeds. This gives the full network of road segments that are accessible under a given scenario from a starting county population center while having consecutive access to a charging station. All consecutively accessible road segments create a cluster of roads that can be reached in an EV without requiring a high degree of planning to avoid running out of battery charge. If a segment is eligible (considered to be covered under the scenario run) by having a charging station, but there is a gap of more than 50 miles between the cluster of accessible roads and the given charging station, the road segment is not considered consecutively accessible. The road segments that are accessible to EV users from a given county feed into the numerator for the coverage metric (Supplementary Fig. 8).

For each county, once the breadth-first search function is run and the cluster of consecutively accessible road segments is found from a given county ($j$), the coverage metric is computed in Equation (1). Each road segment ($i$) in the cluster is weighted by the annual average daily traffic ($w_i$). The consecutive coverage from a starting county ($\rho_j$) is the sum of the set of all consecutively accessible roads ($x_i$) weighted by their annual average daily traffic ($w_i$) over the sum of the set of all roads weighted by their annual average daily traffic, as seen in Equation (1). State coverage ($\rho_s$) is calculated as the average coverage of all counties in the state ($\sigma_s$) weighted by the percent of the state's registered vehicles per county ($v_j$), as seen in Equation (2).

$$\rho_j = \frac{\sum_{i=1}^{I} w_i x_i}{\sum_{i=1}^{I} w_i} \tag{1}$$

$$\rho_s = \frac{\sum_{j=1}^{J} v_j \rho_j}{\sum_{j=1}^{J} v_j} \tag{2}$$

Equation (3) measures the percent of coverage for each county ($j$) from segments that are out of state ($o_i$), therefore measuring the percentage of spillover coverage ($\sigma_j$) from other states. The state-level out-of-state spillover ($\sigma_s$) is calculated as the vehicle-weighted ($v_j$) average out-of-state spillover, as seen in Equation (4).

$$\sigma_j = \frac{\sum_{i=1}^{I} w_i o_i x_i \rho_j}{\sum_{i=1}^{I} x_i w_i} \tag{3}$$

$$\sigma_s = \frac{\sum_{j=1}^{J} v_j \sigma_j}{\sum_{j=1}^{J} v_j} \tag{4}$$

The numerators and denominators in Equations (1) and (3) exclude roads that are less than 10 miles. Short roads are included in the graph data structure and breadth-first search function in case they connect longer roads but are not required to have a charging station to be included in the cluster of segments. Short segments are excluded in the coverage metric because they are primarily in urban areas and would artificially yield a higher number of charging stations required than necessary. If all road segments within 500 miles have a charging station (or are too short to require a charging station), the coverage is 100%.

Our consecutive coverage metric differs from a generic measurement of the total percent of road segments with a charging station by excluding road segments that contain a charging station but are separated by too large of a gap (e.g., EV charging station desert more than 50 miles long) between the starting county and the road segment for an EV to reach it. Chargers past a large gap of charging stations along a road are irrelevant for long-distance trips from a given starting county. We assume that for a long-distance road trip, any road exiting the county's 50-mile population center buffer is accessible from the starting point of the trip, making the consecutive coverage agnostic to the exact starting point within the county center buffer.

### Scenarios

We apply our consecutive coverage metric under three scenarios: Minimum Viable Coverage, Fast Charger Coverage, and AFCs Reach NEVI-Compliant Status. Table 2 shows the parameters for each scenario. The Minimum Viable Coverage scenario considers all planned and existing nonproprietary public Level 2 and DC fast charging stations. The Fast Charger Coverage scenario considers all planned and existing NEVI-compliant charging stations. To be NEVI-compliant, stations must have at least four chargers, which makes them unlikely to have significant wait times. The AFCs Reach NEVI-Compliant Status scenario models coverage of existing and planned NEVI-compliant stations once they are fully built out along all AFCs.

### Road segment definition

Any NHS highway segment longer than 10 miles that does not have a charging station within 1 mile is assumed to require a charging station.

### Table 2 | Scenario description

|  | Minimum Viable Coverage | Fast Charger Coverage | AFCs Reach NEVI-Compliant Status |
|---|---|---|---|
| Charger accessibility | nonproprietary plugs | nonproprietary plugs | nonproprietary plugs |
| Minimum requirement | ≥1 Level 2 & DC fast charger ports | ≥ 4 DC fast charger ports | ≥ 4 DC fast charger ports |
|  | (including < 4) | (NEVI-compliant) | (NEVI program complete) |
| Timeline | Current conditions (2023) | Current conditions (2023) | Future scenario (NEVI program complete) |
| Type of charger | Level 2 & DC fast charger | DC fast charger | DC fast charger |
| Wait time | If long waits to charge are acceptable | Charging experience closer to refueling at gas stations | Charging experience closer to refueling at gas stations |

Outline of the parameters for the three scenarios modeled in this analysis. Alternative Fuel Corridors (AFCs) reach National Electric Vehicle Infrastructure (NEVI) Compliant status represents designated AFCs receiving full coverage of NEVI-compliant direct current (DC) fast-charging stations.

We estimate the charging stations needed by summing the number of segments that lack a charging station within a mile for any given station. This estimate could potentially be optimized (requiring fewer total chargers) by strategically placing charging stations at the intersection of two road segments both lacking a charging station (therefore requiring one charging station to give coverage to two road segments). If a charging station is in the model and falls within 1 mile of multiple roads, both roads are considered to have a charging station; however, to find the number of stations needed, we assume each segment gets a charging station and no further strategic placement occurs.

## Reporting summary

Further information on research design is available in the Nature Portfolio Reporting Summary linked to this article.

## Data availability

The data that support the findings of this study are publicly available. The road data can be found from the National Highway System at the following links https://www.fhwa.dot.gov/planning/national_highway_system/nhs_maps/, and https://www.fhwa.dot.gov/environment/alternative_fuel_corridors/The data regarding the locations of existing and proposed Electric Vehicle charging stations can be found from the U.S. Department of Energy https://afdc.energy.gov/stations/#/analyze?fuel=ELEC&status=E&status=P. The population and vehicle registration data can be found from the U.S. Census Bureau at the following links https://www2.census.gov/geo/docs/reference/cenpop2020/county/CenPop2020_Mean_CO.txt and https://data.census.gov/table/ACSST1Y2021.S2504?q=vehicle. A GitHub repository containing the data used in the analysis is available (https://github.com/Lilyhanig/EV_LongDistance_Coverage). Source data are provided with this paper.

## Code availability

All data analysis was conducted using R. A GitHub repository containing the code used in the analysis is available (https://github.com/Lilyhanig/EV_LongDistance_Coverage). Source data are provided as a Source Data file for state-level figures.

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

## Acknowledgements

This work was authored in part by authors at the National Renewable Energy Laboratory, operated by Alliance for Sustainable Energy, LLC, for the U.S. Department of Energy (DOE) under Contract No. DE-AC36-08GO28308, and in part by authors at Lawrence Berkeley National Laboratory, for the U.S. Department of Energy under Contract No. DE-AC02-05CH11231. Funding provided by the U.S. Department of Energy, Office of Energy Efficiency and Renewable Energy, Vehicle Technologies Office - Energy Efficient Mobility Systems Program. The views expressed in the article do not necessarily represent the views of the DOE or the U.S. Government. The U.S. Government retains and the publisher, by accepting the article for publication, acknowledges that the U.S. Government retains a nonexclusive, paid-up, irrevocable, worldwide license to publish or reproduce the published form of this work, or allow others to do so, for U.S. Government purposes. The authors acknowledge support from the National Science Foundation Graduate Research Fellowship under Grant Numbers DGE1745016 and DGE2140739. We would also like to thank the IN2 2022 Channel Partner Strategic Award for funding and collaboration support. This research was also supported by the Wilton E. Scott Institute for Energy Innovation, Department of Energy Vehicle Technologies Office (Grant No. DEEE0010640), and Alfred P. Sloan Foundation (Grant No. 2024-22563).

## Author contributions

The study was conceived and designed by L.H., C.L., E.W., A.Y., and C.A.S. The data analysis and calculations were conceived and performed by L.H. and C.L., with support from E.W. and D.N. The calculations were analyzed by L.H., C.L., E.W., D.N., C.D.H., and A.Y. The paper was written by L.H., C.L., D.N., and C.D.H., with assistance from E.W., A.Y., and C.A.S.

## Competing interests

The authors declare no competing interests.
