## [Peer Review File · Nature Communications]

REVIEWER COMMENTS

Reviewer #1 (Remarks to the Author):

In this study the authors study the problem of locating charging stations for the purpose of overcoming range anxiety. They propose a new metric called "consecutive coverage" to identify coverage of 500 miles from any given road with a charging station. They use this metric to measure different configurations of planned and existing charging station locations to assess effectiveness of the NEVI compliance. While I appreciate the authors' large scale scope of tackling this problem, there are a number of issues with this work that prevent me from recommending this for publication.

1- In the literature review, the authors seem to suggest they are optimizing charging station locations, but they are not doing that. They are merely computing a metric they invented on planned locations. The metric itself is simply calculating the distance that can be covered on a road network (which has been done in the literature) up to 500 miles, which is an arbitrary distance. The metric itself is also arbitrary, which does not consider distributions of users' travel distances, behavioral range thresholds, or energy consumption patterns, work/home/activity patterns, etc., most of which have been well studied in the literature already. In essence, this study adopts a measure from the literature from 15 years back, presents it as new, and simply uses it without any justification over other metrics in more recent works.

2- So the method is not innovative, but does it provide insights to the NEVI program? The problem is that it does not quantify how effective the NEVI program is, it merely quantifies how effective it is with respect to this invented arbitrary metric. Who is to say if I used a metric of coverage measured by distance weighted by likelihood of trip distance given travel patterns in a region, my metric would not be more informative? If the threshold of 500 miles was changed arbitrarily to 400 miles, how different would the coverage maps look? Ultimately, the effectiveness needs to be validated using data showing that any decisions recommended by the metric (where locations may be lacking) are indeed lacking and vice versa, but no such validation is provided. Instead the authors simply run a number of counterfactual scenarios, which on their own are meaningless without any validation.

3- The authors keep using the term "charging anxiety", but I've only ever heard of "range anxiety" in the literature. Even the proposed consecutive coverage metric (which again is simply a metric of how far one can travel 500 miles out, from the description -- no mathematical or algorithmic description is provided) does not properly capture range anxiety since it ignores other chargers along the way, or queueing, or charging time, etc.

Minor issues:

The authors mix [x] refs with (author,yr).

Reviewer #2 (Remarks to the Author):

See attached review comments in "Nature Communications Review - .pdf"

Reviewer #3 (Remarks to the Author):

This paper introduces a novel metric named "consecutive coverage" to assess the feasibility of long-distance travel across the U.S. states, which in turn provides an indication to the accessibility of EV charging infrastructure. The work done by the authors is interesting and highlights a new perspective of EV charging infrastructure coverage.

Description of the conducted scenarios were more concrete in later sections after the "Scenarios" section. However, since the "Scenarios" section appears first, I recommend that the authors revise it to indicate the characteristics of each scenario clearly.

I also recommend that the authors discuss in the conclusion section how the "consecutive coverage" metric can be generalized to include other weights rather than the average annual daily traffic. For example: congestion level, temperature, ..., etc. This can be a further progression for this metric to compare between EV charging infrastructure coverage in cities with different geographical context.

Reviewer #4 (Remarks to the Author):

Review of “Finding Gaps in US National EV Charging Station Coverage”

Overall Comment

This paper presents a charging station siting optimization model, which is novel and which supports more effective EV adoption in the US. While EV adoption is certainly a tool for improving climate performance, the paper does not directly address environmental impacts, nor does it attempt to evaluate some of the well-known sustainability issues surrounding EV expansion; namely preserving a high-level of travel activity in the US, while not addressing global equity, and the real possibility that widespread electrification will not deliver global carbon emissions reductions, absent an overarching carbon reduction policy or program.

Thus, the paper may fall outside the Aims and Scope of *Nature Communications* given the journal's focus on sciences and applied engineering questions. It falls better into the realm of transportation and infrastructure modeling. Furthermore, no papers from *Nature Communications* are cited in the paper, calling into question the articles fit.

We provide a detailed review of the paper below, with suggestions for supporting the article's linkages to decarbonization and environmental sustainability.

Paper Review

The authors have produced a comprehensive and novel research study that introduces a metric capable of evaluating EV charging station coverage nationwide for long-distance trips. Through the definition of a "consecutive coverage" metric, they identified counties with minimum viable access to EV charging stations and coverage along the Alternative Fuel Corridors (AFC) for potential long-term trips. They found that only 10% of counties in US had minimum viable coverage, which encompasses all charging stations with at least level 2 charger spaced at 50 miles or less with 75% of the highways consecutively covered for long-distance travel. When they looked at if all AFCs have National Electric Vehicle Infrastructure (NEVI) program compliant chargers, 94% of the counties will have consecutive coverage for at least 75% of travel for 500 miles or less travel. Additionally, they did a sensitivity analysis of Tesla's proprietary chargers and how much coverage they will provide to make all AFCs have NEVI compliant chargers and found that they could provide up to \$41 million in saved.

The significance of their work lies in its ability to provide policymakers with insights into the current coverage of EV charging stations and the potential impact of implementing NEVI compliant chargers along AFCs in 50-mile segments. By addressing this gap and evaluating the effectiveness of AFCs in improving national coverage, their research becomes highly valuable for federal and state government agencies.

While this work only addresses a fraction of the broader EV charging infrastructure challenge, its novelty lies in its alignment with the capabilities approach that is gaining traction in the field of transportation. While the study is missing some significant studies on variables that affect EV adoption and has only two prior work that is used as the fundamental building block of the study in terms of charging coverage, the studies referenced are significant in the field. However, I believe the literature review portion can be improved. I attach several studies that could enhance the literature review of this study below.

EV adoption range/charging quality relevant studies:

First of all, some reference/acknowledgement should be given to the collective body of work done by the National Renewable Energy Laboratory (NREL) in analyzing charging station coverage needs across the US. Their EVI-Pro tool, for example, provides decisionmakers with an estimation of plug counts required to meet charging needs at the level of Metropolitan Areas and States. See:

- NREL (National Renewable Energy Laboratory), & DOE (Department of Energy). (2022). Electric Vehicle Infrastructure Projection Tool (EVI-Pro) Lite. Alternative Fuels Data Center. <https://afdc.energy.gov/evi-pro-lite>

Other studies to note on EV adoption range/charging:

- Hidrue, M. K., Parsons, G. R., Kempton, W., & Gardner, M. P. (2011). Willingness to pay for electric vehicles and their attributes. *Resource and energy economics*, 33(3), 686-705.
- Ziefle, M., Beul-Leusmann, S., Kasugai, K., & Schwalm, M. (2014). Public perception and acceptance of electric vehicles: exploring users' perceived benefits and drawbacks. In *Design, User Experience, and Usability. User Experience Design for Everyday Life Applications and Services: Third International Conference, DUXU 2014, Held as Part of HCI International 2014, Heraklion, Crete, Greece, June 22-27, 2014, Proceedings, Part III 3* (pp. 628-639). Springer International Publishing.
- Neves, S. A., Marques, A. C., & Fuinhas, J. A. (2019). Technological progress and other factors behind the adoption of electric vehicles: Empirical evidence for EU countries. *Research in Transportation Economics*, 74, 28-39.
- Higuera-Castillo, E., Guillén, A., Herrera, L. J., & Liébana-Cabanillas, F. (2021). Adoption of electric vehicles: Which factors are really important?. *International Journal of Sustainable Transportation*, 15(10), 799-813.

Charging Infrastructure & environmental impacts:

- Vergis, S., & Chen, B. (2015). Comparison of plug-in electric vehicle adoption in the United States: A state by state approach. *Research in Transportation Economics*, 52, 56-64.
- Mulrow, J., & Grubert, E. (2023). Greenhouse gas emissions embodied in electric vehicle charging infrastructure: A method and case study of Georgia, US 2021–2050. *Environmental Research: Infrastructure and Sustainability*, 3(1), 015013. <https://doi.org/10.1088/2634-4505/acc548>

Additionally, long term trips are more prominent among freight travel rather than personal travel. I see this as a big gap this study easily could have handled but choose not to. Additionally, recent studies showed the importance of reliability of the charging stations for truckers (Konstantinou & Gkritza, 2023) (even if this was not the concern of the study) increases the likelihood of them using this charging infrastructure is higher than personal travel.

- Konstantinou, T., & Gkritza, K. (2023). Examining the barriers to electric truck adoption as a system: A Grey-DEMATEL approach. *Transportation Research Interdisciplinary Perspectives*, 17, 100746.

The analysis and the subsequent figures is sufficient in their takeaways. The takeaways are well-supported with variety of analysis. However, there is potential for further enhancement by incorporating considerations for freight travel or distinguishing between personal and freight travel with dynamic maximum trip distances. While such an addition could bolster the impact of the results, it's acknowledged that the current findings are sufficient, especially given the lack of available datasets for this application.

Another missed opportunity in the study is the analysis of equity implications. While county-specific demographics may not yield significant differences between demographic groups, an examination of equity implications could enhance the overall significance of the research. Testing for equity and other social implications could be incorporated into sensitivity analysis. For example, in Mulrow & Grubert (2023) – referenced above – the authors tested how results would differ if travel behavior (i.e., miles traveled per year) were to shift up or down over time. They showed that a 10% decrease in travel demand would yield greater carbon emissions savings than the studied effort to increase EV adoption through accelerated EV charging deployment. It is important for modeling efforts of this kind to leave room for these kind of social changes, lest we assume that accelerated technology adoption alone can solve for global-scale environmental problems such as climate change. Such a narrow view of technology's ability to yield global reductions in environmental impact has been critiqued elsewhere, for example in:

- Mulrow, J., Derrible, S., & Samaras, C. (2019). Sociotechnical convex hulls and the evolution of transportation activity: A method and application to US travel survey data. *Technological Forecasting and Social Change*, 149, 119789.

- Zoellick, J. C., & Bisht, A. (2018). It's not (all) about efficiency: Powering and organizing technology from a degrowth perspective. *Journal of Cleaner Production*, 197, 1787–1799.

In terms of their methods, the assumptions are appropriate and well-explained. They used established thresholds and common distance buffers in the field of transportation research.

I attached smaller comments regarding style, language, and visual clarity below.

1. Page 3, line 43: "Bipartisan Infrastructure Bill" needs citation.
2. Page 4, line 15: "NHS" is used for the first time without an open-form.
3. Page 4, line 24: The first sentence should be "2022" instead of "2021," as it was 0.5 for 2021 and 0.9 for 2022.
4. Page 4, line 24: While "charging anxiety" is found to be the primary concern in some studies, it is not a "fact." It should be changed to "one of the primary."
5. Page 5, line 16: Mellinger et al. (2018) found that in "Switzerland and Finland," not just Switzerland, as noted in their abstract as "both."
6. Page 5, line 33: This is a repeated sentence from page 5, line 16.
7. Page 6, line 2-11: There needs to be at least one sentence regarding results from both Xie et al. (2018) and Xu et al. (2016). The following paragraph has generalized results, but it will be nice to see the specific results to compare their studies with yours.
8. Page 6, line 26-27: The information inside the parentheses is technically not correct, as better explained on page 27, line 18. It needs to be clarified when the novel metric is first introduced.
9. Page 6, line 42: What does "average annual daily traffic for a road segment" encompass? Personal vehicles? All vehicles?
10. Page 7, line 1: This sentence should not get cut off like this. Consider changing the order of the equation and figure to prevent this.
11. Page 9, line 14: "Out-of-state spillover" needs clarification. Discuss this in the Appendix.
12. Page 16, lines 4-7: I appreciate the discussion on this point.
13. Page 17: Consider mapping changes with and without Tesla chargers for clearer understanding.
14. Page 20, Table 2: Add a line between current conditions (2023) and future scenarios.
15. Page 22, Figure 2 caption, and Page 21, lines 21-22: Ensure consistency in describing colors.

16. Reference number 2: Ensure the citation includes trips above 500 miles. If using NHTS survey data to calculate, the citation should be: U.S. Department of Transportation, Federal Highway Administration, 2022 National Household Travel Survey.
<https://nhts.ornl.gov>
17. Reference number 7: Provide a URL for the reference.
18. References 31, 34, and 35: Check and fix the URL links.
19. Page 27, lines 9-11: For the 31 roads that are AFCs but not in the NHS dataset, discuss why it was not possible to take the average annual traffic from connecting segments that are part of the NHS.

Response to reviewers:

Reviewer #1 (Remarks to the Author):

In this study the authors study the problem of locating charging stations for the purpose of overcoming range anxiety. They propose a new metric called “consecutive coverage” to identify coverage of 500 miles from any given road with a charging station. They use this metric to measure different configurations of planned and existing charging station locations to assess effectiveness of the NEVI compliance. While I appreciate the authors’ large scale scope of tackling this problem, there are a number of issues with this work that prevent me from recommending this for publication.

1. In the literature review, the authors seem to suggest they are optimizing charging station locations, but they are not doing that. They are merely computing a metric they invented on planned locations. The metric itself is simply calculating the distance that can be covered on a road network (which has been done in the literature) up to 500 miles, which is an arbitrary distance.

We thank the reviewer for your feedback and for highlighting the need for clarity on our coverage metric differentiation from previous literature. We quantify the sufficiency of existing and planned coverage rather than optimize new charging station placement. We elaborate on page 4, lines 20-28: “Several papers use optimization to place charging stations. Bräun, et al. (2018) and Jochem, et al. (2019) optimize placement for long-distance travel in Australia and Europe^{[1][2]}. Xie, et al. (2018) use a genetic algorithm to optimize the placement of inter-city chargers in California based on trip origin-destination (OD) pairs^[3]. Xu et al. (2020) optimize placement given demand flow between inter-city OD pairs^[4]. These papers optimize the number of charging stations needed in a network or between OD pairs, but they do not compare station access across different regions, nor do they compute improved access. Instead, we assume consumers need a certain charging station frequency to feel comfortable completing a trip. Our work evaluates the sufficiency of planned or existing charging station infrastructure rather than optimizing infrastructure placement.”

We set the outermost range of distance considered as 500 miles for our metric because less than 0.1% of trips are further than 500 miles according to the Bureau of Transportation Statistics^[5], page 17, line 11:

“The 500-mile range stems from less than 0.1% of all passenger trips being further than 500 miles^[5].”

We focus this analysis on the coverage of long-distance trips due to the focus of the paper assessing coverage improvements from the BIL NEVI program. The goal of the NEVI program is to improve charging station access along alternative fuel corridors for long-distance trips^[6].

Additionally, we have added a sensitivity analysis to test the robustness of the results to the distance threshold. The ‘Distance Sensitivity’ section of the results shows the county coverage metric for 400, 500, and 600 miles. We include the state-level coverage for 400, 500, and 600 miles in the ‘Distance Sensitivity’ Appendix C, pages 29-32.:

Distance Sensitivity

We test the robustness of our coverage metric to the total distance considered in the analysis. Figure 1 shows the minimum viable coverage and fast charger coverage for each county, when looking at a range of 400 miles (Figure 1a and 1b), the baseline case of 500 miles (Figure 1c and 1d), and 600 miles (Figure 1e and 1f). For minimum viable coverage, increasing the range considered shows a similar coverage map, but lowers the magnitude, particularly for urban counties in the center of the US (e.g., Denver, Dallas, Chicago) due to the expanded range including more rural regions with few charging stations. Michigan gained in minimum viable coverage when considering 600 miles due to the expanded range encompassing charging stations on the East Coast. The general pattern of coverage does not shift for

minimum viable coverage or fast charger coverage with West Coast, south Florida, and the Northeast still reaching high coverage, and much of the middle of the country left with low coverage across the distance sensitivity cases.

The fast charger coverage also decreases with increased distance considered, particularly for counties in Colorado that encompass more rural states with an expanded range considered. The general pattern of coverage also does not shift for fast charger coverage, with California, and Washington reaching high coverage across distance scenarios and the remainder of the country reaching coverage below 50%. Table 1 shows the minimum viable coverage, fast charger coverage, and AFC’s reach NEVI-compliance

Figure 1: Map of consecutive coverage at the county level. The top row (a and b) shows maps for coverage within 400 miles of each county population center, the middle row (c and d) shows maps for coverage within 500 miles (the baseline case), and the bottom row shows coverage within 600 miles of each county population center (e and f).

coverage under three sensitivities: a distance range of 400, 500 (the baseline), and 600 miles from each county population center. Coverage tends to decrease slightly as the distance covers more rural ranges. States neighboring high-coverage states tend to gain in coverage when extending the range to 600 miles, such as Oregon (71% to 83% minimum viable coverage). Colorado loses the most coverage when

considering 600 miles compared to 500 miles (from 42% to 25%). Fast Charger coverage is less impacted overall, with Washington losing the most coverage from the 500-mile range to the 600-mile range (49% to 39% fast charger coverage). The AFCs reach NEVI-compliance scenario is largely unaffected by changing the distance to 400 or 600 miles with no state coverage increasing or decreasing by more than 10%. North Dakota gains the most coverage when considering a 600-mile range compared to 500 miles due to its neighboring states' AFC coverage (60% coverage compared to 67%).

Table 1: Table of consecutive coverage under each scenario and averaged for each state by registered vehicle count per county. The out-of-state spillover is the percentage of the 2023 *Minimum Viable Coverage* supplied by other state investments. States are ordered alphabetically.

	Minimum Viable Coverage			Fast Charger Coverage			AFCs Reach NEVI-Compliance		
	400 miles	500 miles	600 miles	400 miles	500 miles	600 miles	400 miles	500 miles	600 miles
Alabama	0.28	0.20	0.14	0.04	0.03	0.02	0.88	0.89	0.90
Alaska	0.46	0.43	0.42	0.40	0.36	0.36	0.70	0.67	0.66
Arizona	0.19	0.16	0.13	0.15	0.12	0.10	0.98	0.98	0.98
Arkansas	0.08	0.05	0.04	0.02	0.02	0.01	0.78	0.79	0.80
California	0.92	0.90	0.84	0.82	0.79	0.71	0.99	0.99	0.99
Colorado	0.62	0.42	0.24	0.35	0.22	0.13	0.89	0.88	0.88
Connecticut	0.92	0.87	0.77	0.39	0.34	0.27	0.98	0.96	0.95
Delaware	0.77	0.68	0.62	0.35	0.28	0.24	0.96	0.95	0.94
D.C.	0.75	0.68	0.58	0.13	0.11	0.09	0.95	0.95	0.94
Florida	0.68	0.50	0.34	0.23	0.17	0.11	0.95	0.93	0.93
Georgia	0.33	0.24	0.17	0.10	0.07	0.05	0.90	0.90	0.91
Idaho	0.16	0.11	0.08	0.05	0.03	0.02	0.95	0.95	0.96
Illinois	0.28	0.22	0.19	0.11	0.08	0.06	0.88	0.87	0.87
Indiana	0.24	0.19	0.19	0.05	0.03	0.02	0.89	0.89	0.90
Iowa	0.27	0.19	0.15	0.03	0.02	0.02	0.83	0.85	0.86
Kansas	0.12	0.07	0.05	0.06	0.03	0.02	0.79	0.82	0.84
Kentucky	0.13	0.10	0.08	0.03	0.02	0.02	0.90	0.90	0.90
Louisiana	0.13	0.08	0.06	0.05	0.03	0.02	0.84	0.85	0.86
Maine	0.95	0.94	0.90	0.05	0.04	0.03	0.98	0.98	0.98
Maryland	0.71	0.65	0.56	0.17	0.14	0.12	0.95	0.95	0.94
Massachusetts	0.99	0.92	0.85	0.26	0.21	0.18	0.98	0.97	0.95
Michigan	0.36	0.42	0.43	0.09	0.05	0.04	0.91	0.91	0.91
Minnesota	0.26	0.17	0.12	0.12	0.08	0.06	0.82	0.84	0.85

Table 1: Table of consecutive coverage under each scenario and averaged for each state by registered vehicle count per county. The out-of-state spillover is the percentage of the 2023 *Minimum Viable Coverage* supplied by other state investments. States are ordered alphabetically.

	Minimum Viable Coverage			Fast Charger Coverage			AFCs Reach NEVI-Compliance		
	400 miles	500 miles	600 miles	400 miles	500 miles	600 miles	400 miles	500 miles	600 miles
Mississippi	0.13	0.08	0.05	0.03	0.02	0.01	0.85	0.86	0.87
Missouri	0.11	0.07	0.05	0.05	0.03	0.02	0.83	0.86	0.87
Montana	0.06	0.03	0.02	0.04	0.02	0.01	0.88	0.90	0.91
Nebraska	0.08	0.05	0.03	0.05	0.03	0.02	0.76	0.82	0.82
Nevada	0.83	0.79	0.72	0.75	0.71	0.63	0.99	0.99	0.98
New Hampshire	0.99	0.95	0.87	0.17	0.14	0.12	0.98	0.98	0.93
New Jersey	0.86	0.77	0.68	0.37	0.29	0.25	0.97	0.96	0.95
New Mexico	0.28	0.25	0.15	0.06	0.04	0.02	0.89	0.89	0.88
New York	0.91	0.78	0.70	0.21	0.17	0.14	0.97	0.95	0.95
North Carolina	0.30	0.21	0.16	0.07	0.04	0.03	0.92	0.93	0.94
North Dakota	0.33	0.21	0.12	0.04	0.02	0.01	0.55	0.60	0.67
Ohio	0.46	0.42	0.39	0.05	0.04	0.03	0.93	0.92	0.92
Oklahoma	0.42	0.31	0.23	0.06	0.04	0.03	0.80	0.82	0.83
Oregon	0.76	0.71	0.83	0.30	0.20	0.15	0.97	0.98	0.98
Pennsylvania	0.76	0.66	0.58	0.25	0.20	0.17	0.95	0.95	0.94
Rhode Island	0.98	0.90	0.80	0.45	0.39	0.32	0.98	0.97	0.92
South Carolina	0.40	0.29	0.21	0.08	0.05	0.04	0.91	0.92	0.93
South Dakota	0.06	0.03	0.02	0.02	0.01	0.01	0.68	0.74	0.76
Tennessee	0.27	0.18	0.13	0.04	0.03	0.02	0.88	0.90	0.90
Texas	0.37	0.30	0.24	0.15	0.11	0.09	0.80	0.79	0.80
Utah	0.56	0.40	0.20	0.16	0.10	0.04	0.94	0.93	0.97
Vermont	0.99	0.96	0.88	0.05	0.04	0.03	0.98	0.97	0.96
Virginia	0.66	0.59	0.52	0.10	0.08	0.06	0.94	0.94	0.94
Washington	0.76	0.69	0.61	0.55	0.49	0.39	0.96	0.96	0.96
West Virginia	0.42	0.36	0.32	0.04	0.03	0.02	0.94	0.94	0.93
Wisconsin	0.28	0.21	0.17	0.10	0.08	0.06	0.88	0.88	0.87
Wyoming	0.36	0.25	0.17	0.12	0.08	0.05	0.90	0.88	0.87

2. The metric itself is also arbitrary, which does not consider distributions of users' travel distances, behavioral range thresholds, or energy consumption patterns, work/home/activity patterns, etc., most of which have been well studied in the literature already.

We focus this analysis on the coverage of long-distance trips due to the focus of the paper assessing coverage improvements from the BIL NEVI program. The goal of the NEVI program is to improve charging station access along alternative fuel corridors for long-distance trips⁶. Additionally, the literature shows the ability to traverse long distances is a priority for consumers^{7,8}. Work/home/activity patterns for day-to-day life may play an important role in siting charging stations for community-level charging, however, these characteristics are outside the scope of relevance for long-distance trips and our analysis. Energy consumption patterns and hourly travel patterns may pose an important problem for EV charging stations on long-distance trips as more consumers adopt EVs and charging stations begin to be crowded. We highlight this as an area for future work in our Appendix A: Limitations and Further Research on page 26, lines 21-29:

“Future research or extensions of this work could calculate the consecutive coverage metric, weighting road segments by peak traffic per year or congestion levels. Another extension of this paper could dynamically define the minimum number of chargers per station needed by the annual average daily traffic or peak daily traffic of the year (i.e., requiring more charging ports for high-trafficked roads). Therefore, not only identifying gaps in coverage, but segments of road that contain an insufficient number of chargers per station. As more consumers adopt electric vehicles, queuing times and high-congestion travel days may become a primary form of charging anxiety. Therefore, extending this work to look specifically at peak congestion both for weighting the road segments in the coverage metric and for considering the sufficiency of chargers per station would be a valuable extension of this work.”

In essence, this study adopts a measure from the literature from 15 years back, presents it as new, and simply uses it without any justification over other metrics in more recent works.

The authors are not aware of a similar metric for charging station coverage. Additionally, fifteen years ago, there were 484 charging stations in the U.S. (with 411 in California alone) compared to the 60,000 charging stations today and the 110,000 gas stations⁹. Therefore, the value of our metric is in its ability to quantify the progress of charging station development, compare long-distance access by location, and show which areas benefit the most from federal spending.

3. So the method is not innovative, but does it provide insights to the NEVI program? The problem is that it does not quantify how effective the NEVI program is, it merely quantifies how effective it is with respect to this invented arbitrary metric. Who is to say if I used a metric of coverage measured by distance weighted by likelihood of trip distance given travel patterns in a region, my metric would not be more informative? If the threshold of 500 miles was changed arbitrarily to 400 miles, how different would the coverage maps look? Ultimately, the effectiveness needs to be validated using data showing that any decisions recommended by the metric (where locations may be lacking) are indeed lacking and vice versa, but no such validation is provided. Instead the authors simply run a number of counterfactual scenarios, which on their own are meaningless without any validation.

Thank you for your feedback. We have added a validation section under results titled ‘Distance Sensitivity’ to show the minimum viable coverage and fast charger coverage county maps for a maximum distance of 400, 500, and 600 miles (shown above). We also have a ‘Distance Sensitivity’ in Appendix B with state-level coverage tables (shown above). We find that changing the distance threshold does not change the results of state or county coverage relative to each other and does not significantly change the magnitude of the results. The

state with the largest change in coverage from using a range of 600 miles instead of 500 miles is Colorado with a 17% difference across the sensitivities.

4. The authors keep using the term “charging anxiety”, but I’ve only ever heard of “range anxiety” in the literature.

It’s true; “charging anxiety” is a new term and may not be well established in the literature yet. We see it as a better-fit word for the evolving barriers to EV adoption and utilization. Therefore, we contribute to the literature by using the term highlighted by Scott Hardman and discussed in the article “Forget Range Anxiety. Most EV Drivers Have Charge Anxiety.”^[10] The term is gaining momentum in the literature with Liu et al. (2023) writing, “The second reason we guessed for consumers’ range anxiety is charging anxiety. It is mainly due to the imperfect charging infrastructure, consumers’ inability to find charging piles in time, or charging time being too long, especially when battery charging is not on the consumers’ schedule.”^[11] Thank you for highlighting the need to clarify our choice of wording. We have added the below text to our introduction on page 2, lines 5-9:

“The literature frequently discusses ‘range anxiety’^{[12][13]}, which implies concerns over the distance an EV can cover from a single charge. Instead, we use the term ‘charging anxiety’ to move from the vehicle-centric ‘range anxiety’ to the infrastructure-centric. Charging anxiety encompasses the sufficiency of EV charging infrastructure to meet changing needs in frequency (gaps in coverage), density (queuing), and reliability (out-of-service).”

Even the proposed consecutive coverage metric (which again is simply a metric of how far one can travel 500 miles out, from the description – no mathematical or algorithmic description is provided) does not properly capture range anxiety since it ignores other chargers along the way, or queuing, or charging time, etc.

We highlight the definition and mathematical equations used in the sections below on Pages 18 and 19:

“For each county, once the breadth-first search function is run and the cluster of consecutively EV-charger accessible road segments are found from a given county (j), the coverage metric is computed (Equation 1). Each road segment (i) is weighted by the Annual Average Daily Traffic (w_i); a binary indicator variable (x_i) denotes if each segment (i) is accessible and therefore included in the set. The consecutive coverage from a starting county (ρ_j) is the sum of the set of all consecutively accessible roads (x_i) weighted by their annual average daily traffic (w_i) over the sum of the set of all roads weighted by their annual average daily traffic, as seen in Equation 1. State coverage (ρ_s) is calculated as the average coverage of all counties in the state (σ_s) weighted by the percent of the state’s registered vehicles per county (v_j), as seen in Equation 2.

$$\rho_j = \frac{\sum_{i=1}^I w_i x_i}{\sum_{i=1}^I w_i} \tag{1}$$

$$\rho_s = \frac{\sum_{j=1}^J v_j \rho_j}{\sum_{j=1}^J v_j} \tag{2}$$

”

Additionally, we have equations in our Methods section on Pages 18-20, subsection ‘Consecutive Coverage Metric Formulation’:

Consecutive Coverage Metric Formulation

The process we take to calculate consecutive coverage for each county is illustrated in figure 2. We consider all three scenarios defined in Table 2 in this analysis.

For each scenario and each county, we create a graph data structure out of all eligible segments, we then run a breadth-first search function outward using all road segments within 50 miles of the county population center as starting seeds. This gives the full network of road segments that are accessible under a given scenario from a starting county population center while having consecutive access to a charging station. All consecutively accessible road segments create a cluster of roads that can be reached in an EV without requiring a high-degree of planning to avoid running out of battery. If a segment is eligible (considered to be covered under the scenario run) by having a charging station, but there is a gap of more 50 miles between the cluster of accessible roads and the given charging station, the road segment is not considered consecutively accessible. The road segments that are accessible (to EV users) from a given county feed into the numerator for the coverage metric as seen in Figure 2

Figure 2: Flow chart of methodology for coverage metric calculation with datasets used in blue, input parameters and assumptions in green, analysis methods in purple, and the final results outputs in blue.

For each county, once the breadth-first search function is run and the cluster of consecutively accessible road segments are found from a given county (j), the coverage metric is computed (Equation 3). Each road segment (i) in the cluster is weighted by the Annual Average Daily Traffic (w_i). The consecutive coverage from a starting county (ρ_j) is the sum of the set of all consecutively accessible roads (x_i) weighted by their annual average daily traffic (w_i) over the sum of the set of all roads weighted by their annual average daily traffic, as seen in Equation 3. State coverage (ρ_s) is calculated as the average coverage of all counties in the state (σ_s) weighted by the percent of the state's registered vehicles per county (v_j), as seen in Equation 4.

$$\rho_j = \frac{\sum_{i=1}^I w_i x_i}{\sum_{i=1}^I w_i} \quad (3)$$

$$\rho_s = \frac{\sum_{j=1}^J v_j \rho_j}{\sum_{j=1}^J v_j} \quad (4)$$

Equation 5 measures the percent of coverage for each county (j) from segments that are out-of-state (o_i); therefore, measuring the percentage of spillover coverage (σ_j) from other states. The state-level out-of-state

spillover (σ_s) is calculated as the vehicle-weighted (v_j) average out-of-state spillover, as seen in Equation 6.

$$\sigma_j = \frac{\sum_{i=1}^I w_i o_i x_i \rho_j}{\sum_{i=1}^I x_i w_i} \quad (5)$$

$$\sigma_s = \frac{\sum_{j=1}^J v_j \sigma_j}{\sum_{j=1}^J v_j} \quad (6)$$

Both the numerator and denominator in Equations 3 & 5 exclude roads that are less than 10 miles in total length. Short roads are included in the graph data structure and breadth-first search function in case they connect longer roads but are not required to have a charging station to be included in the cluster of segments. Short segments are only excluded in the coverage metric because they are primarily in urban areas and would artificially yield a higher number of charging stations required than necessary. If all road segments within 500 miles have a charging station (or were too short to require a charging station), the coverage is 100%.

Our consecutive coverage metric differs from a generic measurement of the total percent of road segments with a charging station by excluding road segments that contain a charging station but are separated by too large of a gap (e.g., EV charging station desert more than 50 miles long) between the starting county and the road-segment for an EV to reach it. Chargers past a large gap of charging stations along a road are irrelevant for long-distance trips from a given starting county. We assume that for a long-distance road trip, any road exiting the county’s 50-mile population center buffer is accessible from the starting point of the trip, making the consecutive coverage agnostic to the exact starting point within the county center buffer.”

[it] does not properly capture range anxiety since it ignores other chargers along the way, or queuing, or charging time, etc.

Our fast charger scenario only considers charging stations with at least four DC fast chargers, which should minimize queuing and wait times in the near future while EV penetration is low. However, we highlight this as an area for future work in our Appendix A, ‘Limitations and Further Research’ on page 26, lines 21-29:

“Future research or extensions of this work could calculate the consecutive coverage metric, weighting road segments by peak traffic per year or congestion levels. Another extension of this paper could dynamically define the minimum number of chargers per station needed by the annual average daily traffic or peak daily traffic of the year (i.e., requiring more charging ports for high-trafficked roads). Therefore, not only identifying gaps in coverage, but segments of road that contain an insufficient number of chargers per station. As more consumers adopt electric vehicles, queuing times and high-congestion travel days may become a primary form of charging anxiety. Therefore, extending this work to look specifically at peak congestion both for weighting the road segments in the coverage metric and for considering the sufficiency of chargers per station would be a valuable extension of this work.”

Minor issues: The authors mix [x] refs with (author,yr).

We have updated the manuscript and cite papers in the following format; standard for Nature Communications papers: “Author (year) discuss xyz¹.”

Reviewer #2:

This paper presents a charging station siting optimization model, which is novel and which supports more effective EV adoption in the US. While EV adoption is certainly a tool for improving climate performance, the paper does not directly address environmental impacts, nor does it attempt to evaluate some of the well-known sustainability issues surrounding EV expansion; namely preserving a high-level of travel activity in the US, while not addressing global equity, and the real possibility that widespread electrification will not deliver global carbon emissions reductions, absent an overarching carbon reduction policy or program. Thus, the paper may fall outside the Aims and Scope of Nature Communications given the journal’s focus on

sciences and applied engineering questions. It falls better into the realm of transportation and infrastructure modeling. Furthermore, no papers from Nature Communications are cited in the paper, calling into question the articles fit.

We provide a detailed review of the paper below, with suggestions for supporting the article’s linkages to decarbonization and environmental sustainability.

We thank the reviewer for highlighting “the real possibility that widespread electrification will not deliver global carbon emissions reductions”. We have expanded on the other paths to decarbonization for the transportation sector beyond technology adoption on Page 5, lines 34-43:

“It is imperative to decarbonize the light-duty vehicle sector to meet climate goals^{[14][15]}. Hoene et al. (2023) outline pathways to decarbonize U.S. passenger and freight vehicles using TEMPO for EV adoption and associated emissions under varying assumptions. Charging station infrastructure is an important driver of EV adoption^[16], but is just one tool available for decarbonizing transportation^{[17][18][19]} Mulrow and Grubert (2023) highlight the potential impact of changing behavior, such as decreasing total miles traveled on overall vehicle emissions^[17]. Aguilar et al. find that if 50% of EVs in Europe implement vehicle-to-grid, they can meet the full demand for battery storage in Europe, reducing infrastructure build-out^[20]. Ren et al. (2023) find that although EVs are lower-emitting than internal combustion engine vehicles over the vehicle lifetime, their greenhouse gas emissions are front-loaded due to battery manufacturing^[21]. Therefore, efforts should strategically replace high-emitting and high-mileage internal combustion engine vehicles with EVs^[21].”

Additionally, we discuss and cite four relevant Nature Communications papers:

Page 4, lines 12-14:

“Lanz et al. (2022) find that the levelized cost of electricity of public chargers in Europe decreases as charging station utilization rates increase, further demonstrating the need for EV infrastructure co-evolution^[22].”

Page 5, lines 40-42:

“Ren et al. (2023) find that although EVs are lower-emitting than internal combustion engine vehicles over the vehicle lifetime, their greenhouse gas emissions are front-loaded due to battery manufacturing^[21].”

Page 5, lines 26-30:

“TEMPO is an important model for assessing the potential for charging station coverage, such as our consecutive coverage metric, to find the change in vehicle adoption and demand due to infrastructure improvements^{[23][24]}. Hoene et al. (2023) use TEMPO to project passenger and freight decarbonization and model the emissions outcomes under varying scenarios such as tightened fuel standards, zero emissions vehicle mandates, and lowering miles traveled^[24].”

and Page 5, lines 39-40:

“Aguilar et al. find that if 50% of EVs in Europe implement vehicle-to-grid, they can meet the full demand for battery storage in Europe, reducing infrastructure build-out^[20].”

Paper Review The authors have produced a comprehensive and novel research study that introduces a metric capable of evaluating EV charging station coverage nationwide for long-distance trips. Through the definition of a “consecutive coverage” metric, they identified counties with minimum viable access to EV charging stations and coverage along the Alternative Fuel Corridors (AFC) for potential long-term trips. They found that only 10% of counties in US had minimum viable coverage, which encompasses all charging stations with at least level 2 charger spaced at 50 miles or less with 75% of the highways consecutively covered for long-distance travel. When they looked at if all AFCs have National Electric Vehicle Infrastructure (NEVI) program-compliant chargers, 94% of the counties will have consecutive coverage for at least 75% of travel for 500 miles or less travel. Additionally, they did a sensitivity analysis of Tesla’s proprietary chargers and how much coverage they will provide to make all AFCs have NEVI compliant chargers and found that

they could provide up to \$41 million in saved.

The significance of their work lies in its ability to provide policymakers with insights into the current coverage of EV charging stations and the potential impact of implementing NEVI-compliant chargers along AFCs in 50-mile segments. By addressing this gap and evaluating the effectiveness of AFCs in improving national coverage, their research becomes highly valuable for federal and state government agencies. While this work only addresses a fraction of the broader EV charging infrastructure challenge, its novelty lies in its alignment with the capabilities approach that is gaining traction in the field of transportation. While the study is missing some significant studies on variables that affect EV adoption and has only two prior work that is used as the fundamental building block of the study in terms of charging coverage, the studies referenced are significant in the field. However, I believe the literature review portion can be improved. I attach several studies that could enhance the literature review of this study below.

EV adoption range/charging quality relevant studies: First of all, some reference/acknowledgement should be given to the collective body of work done by the National Renewable Energy Laboratory (NREL) in analyzing charging station coverage needs across the US. Their EVI-Pro tool, for example, provides decisionmakers with an estimation of plug counts required to meet charging needs at the level of Metropolitan Areas and States. See:

- We expand on NREL and EVI-Pro’s contributions on page 5, lines 10-33:

“NREL has several tools for assessing EV charging station needs in communities and for long-distance trips. The Electric Vehicle Infrastructure Program (EVI-Pro) model can be used to find the total charging stations needed for a metropolitan statistical area and the associated electricity demand²⁵. The set of EVI-X modeling tools helps address EV charging station questions from different angles, such as charging station need at the community level (EVI-Pro), siting charging stations for long-distance trips (EVI-RoadTrip) and planning charging station design²⁶. Wood et al. (2023) use the EVI-X suite of models to find the build-out of charging station equipment needed to meet forecasted EV adoption out to 2030²⁷. EVI-Pro estimates the number of charging stations needed in a metropolitan area to meet community-level charging demand, while our metric assesses the ability to drive long distances without hitting gaps in coverage. Additionally, our metric is at the county level for all U.S. counties, and EVI-Pro lite is limited to metropolitan statistical areas. EVI-RoadTrip is a useful model for assessing charging station adequacy for a specific route²⁶. In contrast, our metric shows regional charging station adequacy and is useful for comparing across counties, states, and policies.

NREL’s Transportation Energy & Mobility Pathway Options Model (TEMPO), is an energy systems model of the U.S. transportation system²³. Among other features, TEMPO estimates vehicle stock, new technology adoption (including types of EVs), activity, and energy consumption for the entire U.S. light-duty vehicle fleet²³. TEMPO is an important model for assessing the potential for charging station coverage, such as our consecutive coverage metric, to find the change in vehicle adoption and demand due to infrastructure improvements^{23/24}. Hoene et al. (2023) use TEMPO to project passenger and freight decarbonization and model the emissions outcomes under varying scenarios such as tightened fuel standards, zero emissions vehicle mandates, and lowering miles traveled²⁴. The model considers charging station access when modeling EV adoption propensity and associated changes in charging load. Therefore, our metric could be used with a model such as TEMPO to inform their adoption propensity with a more nuanced assessment of long-distance charging station coverage.”

- Other studies to note on EV adoption range/charging:

Hidrué, M. K., Parsons, G. R., Kempton, W., Gardner, M. P. (2011). Willingness to pay for electric vehicles and their attributes. *Resource and energy economics*, 33(3), 686- 705.

We have added the above reference to our discussion in the literature review on Page 4, lines 34-36:

“ Hidrué et al. (2011) look at the value of extending EV battery range to consumers, finding that survey respondents were willing to pay up to \$75 per additional mile of driving range, with a decreased willingness to pay as the range increases⁷.”

- Ziefle, M., Beul-Leusmann, S., Kasugai, K., & Schwalm, M. (2014). Public perception and acceptance

of electric vehicles: exploring users' perceived benefits and drawbacks. In Design, User Experience, and Usability. User Experience Design for Everyday Life Applications and Services: Third International Conference, DUXU 2014, Held as Part of HCI International 2014, Heraklion, Crete, Greece, June 22-27, 2014, Proceedings, Part III 3 (pp. 628-639). Springer International Publishing.

We have added the above citation to our literature review on Page 3 lines 39-41:

“Additionally, Ziefle et al. (2014) found in a survey that the largest two barriers to EVs were comfort and technology, defined as comfort in re-fueling an EV and availability of charging stations^[28]. ”

- Neves, S. A., Marques, A. C., & Fuinhas, J. A. (2019). Technological progress and other factors behind the adoption of electric vehicles: Empirical evidence for EU countries. *Research in Transportation Economics*, 74, 28-39.

We have cited the above paper on Page 4, lines 4-6:

“Almeida Neves (2019) et al. use regression to study the factors influencing battery and plug-in electric vehicle adoption in 24 EU countries, finding that access to charging stations drive adoption across types of EVs ”

- Higuera-Castillo, E., Guillén, A., Herrera, L. J., & Liébana-Cabanillas, F. (2021). Adoption of electric vehicles: Which factors are really important?. *International Journal of Sustainable Transportation*, 15(10), 799-813.

We have cited the above paper on Page 3, lines 41-42:

“Similarly, Higuera-Castillo et al. (2021) find range and reliability are top predictors of intention to purchase an EV among potential consumers surveyed in Spain^[29]. ”

Charging Infrastructure & environmental impacts:

- Vergis, S., & Chen, B. (2015). Comparison of plug-in electric vehicle adoption in the United States: A state by state approach. *Research in Transportation Economics*, 52, 56- 64.

We have cited the above paper on Page 4, lines 6-8:

“Comparatively, Vergis and Chen (2015) find access to charging stations to be a leading predictor of battery EV adoption, but a less significant factor for plug-in hybrid EV adoption^[30]. ”

- Mulrow, J., & Grubert, E. (2023). Greenhouse gas emissions embodied in electric vehicle charging infrastructure: A method and case study of Georgia, US 2021–2050. *Environmental Research: Infrastructure and Sustainability*, 3(1), 015013. <https://doi.org/10.1088/2634-4505/acc548>

We cite the above paper on Page 5, lines 36-39:

“Charging station infrastructure is an important driver of EV adoption^[16], but is just one tool available for decarbonizing transportation^{[17][18][19]} Mulrow and Grubert (2023) highlight the potential impact of changing behavior, such as decreasing total miles traveled on overall vehicle emissions^[17]. ”

Additionally, long term trips are more prominent among freight travel rather than personal travel. I see this as a big gap this study easily could have handled but choose not to. Additionally, recent

studies showed the importance of reliability of the charging stations for truckers (Konstantinou Gkritza, 2023) (even if this was not the concern of the study) increases the likelihood of them using this charging infrastructure is higher than personal travel.

We have added Appendix B assessing the consecutive coverage of electric vehicle charging stations for long-distance trips of both medium and heavy-duty trucks on Pages 27-28:

0.1 Medium and Heavy-Duty Truck Coverage

Medium and heavy-duty trucks have different vehicle sizes and trailer requirements from light-duty vehicles, requiring larger station designs³¹ and in some cases higher-powered charging³². There are 400 medium-duty (Class 3-5, or a gross vehicle weight rating [GVWR] of 10,001 to 19,500 pounds) and 100 heavy-duty (Class 6-8, or a GVWR of 19,501 pounds and above) public electric vehicle charging stations in the US as of April 2024 shown in Figure 3, according to the Alternative Fuels Data Center⁹, compared to the 62,000 public charging stations for light-duty vehicles. We exclude private depot chargers. Figures 4 and 5 show the county-level minimum viable consecutive coverage for medium-duty and heavy-duty trucks, respectively. California reaches the highest state-level charging station coverage for both medium and heavy-duty coverage (24% in both cases), while 37 states have state-level coverage below 10% for medium-duty and 40 states have coverage below 10% for heavy-duty. Charging station access and investment for freight lags significantly behind that of light-duty vehicles.

Figure 3: Public Electric Vehicle chargers for medium and heavy-duty trucks in the US as of April 2024. The orange dots show all stations that have a maximum vehicle class of Heavy-Duty (allowing for medium and heavy-duty). The dark green dots show all stations with a maximum medium-duty vehicle class. Charging station data was retrieved April 9st, 2024.

Figure 4: Charging station consecutive coverage of medium-duty trucks for all level 2 and DC fast chargers within 1 mile of a National Highway System road, *Minimum Viable Coverage Scenario*.

Figure 5: Charging station consecutive coverage of medium and heavy-duty trucks for all level 2 and DC fast chargers within 1 mile of a National Highway System road, *Minimum Viable Coverage Scenario*.

The analysis and the subsequent figures is sufficient in their takeaways. The takeaways are well-supported with variety of analysis. However, there is potential for further enhancement by incorporating considerations for freight travel or distinguishing between personal and freight travel with dynamic maximum trip distances. While such an addition could bolster the impact of the results, it's acknowledged that the current findings are sufficient, especially given the lack of available datasets for this application.

We have added maps and a discussion of consecutive coverage access for medium and heavy-duty trucks on Pages 27-28 (shown above).

Another missed opportunity in the study is the analysis of equity implications. While county-specific demographics may not yield significant differences between demographic groups, an examination of equity implications could enhance the overall significance of the research. Testing for equity and other social implications could be incorporated into sensitivity analysis.

We have added a section, Appendix E: 'Equity Implications', looking at consecutive coverage of charging stations compared to median household income and percentage of car dependence among commuters per county on Page 35:

“Figure 6 shows the median house income per county from the American Community Survey for 2022 compared against the state of coverage under minimum viable coverage, fast-charger coverage, and when AFCs reach NEVI-compliant status with the color representing the percentage of residents that commute by car. The U.S. is highly car-dependent; the mean percentage of residents commuting by car is 87%, according to the American Community Survey³³. The current state of minimum viable coverage in the U.S. trends slightly higher with median household income. Very urban counties with commuter car dependence below 25% have long-distance minimum viable charging station coverage at or above 50%; minimum viable coverage is higher among urban counties despite lower car-dependence overall in cities, leaving the most car-dependent and low household income counties largely with lower coverage overall. The same trend continues when looking at fast charger coverage, with only two counties below the average household income threshold having fast charger coverage above 50%. When AFCs reach NEVI-compliant status, the majority of counties (94%) reach coverage above 10%. The counties left with coverage below 10% are very rural and represent a range of county-level median household incomes but are primarily car-dependent.”

Figure 6: Median household income of each county in the US versus charging station coverage for minimum viable coverage, fast charger coverage, and when AFCs reach NEVI-compliant status. The color represents the percentage of residents in each county that commute by car, van, or light-duty truck.

We discuss on Page 14, lines 11-16, how the counties left at very low coverage are also rural counties, a dimension of equity within the context of infrastructure:

“For all counties to reach 100% fast charger coverage beyond AFCs, 4,500 additional road segments need NEVI-compliant charging stations (four universal-plug DC fast chargers), primarily in rural counties; this gap between full coverage of AFCs and full coverage on NHS roads represents 24% of NHS traffic. This requires 4500 additional charging stations because extending coverage into rural areas provides less-traffic weighted coverage. AFCs comprise interstates and other high-traffic roads, which are low-hanging fruit for reaching high consecutive charging station coverage.”

For example, in Mulrow & Grubert (2023) – referenced above – the authors tested how results would differ if travel behavior (i.e., miles traveled per year) were to shift up or down over time. They showed that a 10% decrease in travel demand would yield greater carbon emissions savings than the studied effort to increase EV adoption through accelerated EV charging deployment. It is important for modeling efforts of this kind to leave room for these kind of social changes, lest we assume that accelerated technology adoption alone can solve for global-scale environmental problems such as climate change. Such a narrow view of technology’s ability to yield global reductions in environmental impact has been critiqued elsewhere, for example in:

- Mulrow, J., Derrible, S., Samaras, C. (2019). Sociotechnical convex hulls and the evolution of transportation activity: A method and application to US travel survey data. *Technological Forecasting and Social Change*, 149, 119789.
- Zoellick, J. C., Bisht, A. (2018). It’s not (all) about efficiency: Powering and organizing technology from a degrowth perspective. *Journal of Cleaner Production*, 197, 1787–1799.

We have expanded on the potential for both degrowth and lower vehicle miles traveled to contribute

to decarbonization on page 5, lines 34-43, and highlighted below:

“It is imperative to decarbonize the light-duty vehicle sector to meet climate goals^{[14][15]}. Hoene et al. (2023) outline pathways to decarbonize U.S. passenger and freight vehicles using TEMPO for EV adoption and associated emissions under varying assumptions. Charging station infrastructure is an important driver of EV adoption^[16], but is just one tool available for decarbonizing transportation^{[17][18][19]}. Mulrow and Grubert (2023) highlight the potential impact of changing behavior, such as decreasing total miles traveled on overall vehicle emissions^[17]. Aguilar et al. find that if 50% of EVs in Europe implement vehicle-to-grid, they can meet the full demand for battery storage in Europe, reducing infrastructure build-out^[20]. Ren et al. (2023) find that although EVs are lower-emitting than internal combustion engine vehicles over the vehicle lifetime, their greenhouse gas emissions are front-loaded due to battery manufacturing^[21]. Therefore, efforts should strategically replace high-emitting and high-mileage internal combustion engine vehicles with EVs^[21].”

In terms of their methods, the assumptions are appropriate and well-explained. They used established thresholds and common distance buffers in the field of transportation research. I attached smaller comments regarding style, language, and visual clarity below.

1. Page 3, line 43: “Bipartisan Infrastructure Bill” needs citation.

Page 2, lines 34-38: Two citations have been added. It now reads:

“The federal government committed to investing up to \$7.5 billion into public EV charging infrastructure through the Bipartisan Infrastructure Law (BIL)^{[6][34]}. This consists of \$5.0 billion for the National Electric Vehicle Infrastructure (NEVI) Formula Program administered by the U.S. Department of Transportation through the states and \$ 2.5 billion for the Charging and Fueling Infrastructure Discretionary Grant Program^[6]. Additionally, \$3.0 billion in public investment has been made across all levels of government, led by programs from California^[6].”

2. Page 4, line 15: “NHS” is used for the first time without an open-form.

On Page 3, line 10-12, the open form, National Highway System, has been added:

“We create a novel ‘consecutive coverage’ metric, which measures the percent of National Highway System (NHS) roads (traffic-weighted) that are consecutively accessible within 500 miles of each starting county.”

3. Page 4, line 24: The first sentence should be “2022” instead of “2021,” as it was 0.5 for f2021 and 0.9 for 2022. The sentence has been removed due to limited space in Nature Communications publications and lack of relevancy.

4. Page 4, line 24: While “charging anxiety” is found to be the primary concern in some studies, it is not a “fact.” It should be changed to “one of the primary.”

Page 3, line 21: The language has been changed to: “Charging anxiety (sometimes referred to as ‘range anxiety’) is one of the primary concerns among consumers^{[8][35][12][36]}.”

5. Page 5, line 16: Mellinger et al. (2018) found that in “Switzerland and Finland,” not just Switzerland, as noted in their abstract as “both.”

6. Page 5, line 33: This is a repeated sentence from page 5, line 16.

Page 4, line 41-43: The sentence has been corrected to include Switzerland and Finland, and the second instance has been removed to avoid redundancy:

“Mellinger et al. (2018) show that in Finland and Switzerland, more public charging infrastructure near homes would allow 99% of trips to be completed in an EV^[13]. They define coverage by the percentage of annual trips that can be completed in an EV. ” It has also been moved down to replace line 33 to avoid repetition.

7. Page 6, line 2-11: There needs to be at least one sentence regarding results from both Xie et al. (2018) and Xu et al. (2016). The following paragraph has generalized results, but it will be nice to see the specific results to compare their studies with yours.

Both papers provide insightful optimization formulations for siting EV charging stations under their own formulation of coverage: percentage of viable inter-city OD pair trips in Xie et al. (2018) and the total range anxiety as a function of state-of-charge within a simulation in Xu et al. (2020). However, neither paper shows the current state of coverage and in particular, the Xu et al. paper range-anxiety metric makes comparisons to this paper challenging. Xie, et al. (2018) use a genetic algorithm is used to optimize the placement and sequencing of inter-city chargers in California given sets of trip origin-destination (OD) pairs^[3]. This analysis is only conducted within California, so conclusions cannot be drawn regarding the viability of inter-state travel. Additionally, they define coverage as the percentage of OD pair trips that can be completed given charging station placement. It does not compare coverage across different regions or give a sense of the equity distribution of charging station access in their optimization^[3]. Xu et al. (2020) place charging stations in an optimization given demand flow between origin-destination pairs for inter-city travel with a charging-anxiety constraint assuming non-linear charging anxiety with percent battery capacity remaining. They use the Texas highway system (124 nodes, 238 edges) as their case study. This study provides insights into the optimal placement of charging stations when minimizing range anxiety (as defined by a non-linear relationship to state-of-charge)^[4], however, it does not directly yield a comparison of the current state of charging station coverage by region to compare distribution.

We have modified the language on Page 4 lines 20-28:

“Several papers use optimization to place charging stations. Bräun, et al. (2018) and Jochem, et al. (2019) optimize placement for long-distance travel in Australia and Europe^[12]. Xie, et al. (2018) use a genetic algorithm to optimize the placement of inter-city chargers in California based on trip origin-destination (OD) pairs^[3]. Xu et al. (2020) optimize placement given demand flow between inter-city OD pairs^[4]. These papers optimize the number of charging stations needed in a network or between OD pairs, but they do not compare station access across different regions, nor do they compute improved access. Instead, we assume consumers need a certain charging station frequency to feel comfortable completing a trip. Our work evaluates the sufficiency of planned or existing charging station infrastructure rather than optimizing infrastructure placement.”

8. Page 6, line 26-27: The information inside the parentheses is technically not correct, as better explained on page 27, line 18. It needs to be clarified when the novel metric is first introduced.

We have clarified the language on page 6, lines 3-9, to better explain our coverage metric per the appendix:

“We use a breadth-first search function: an algorithm that searches along a tree data structure, (e.g., a network of road segments) for a charging station and continues to search along all connected segments that contain a charging station. This provides a comparison of long-distance traversability for EVs from every county in the US (within 500 miles of each county population center and outside of a 50-mile buffer of each county population center). We use geospatial highway data coupled with public EV charging station location data in the breadth-first search to find the percent of highways (traffic-weighted) that are consecutively accessible to public charging stations from a given starting county without gaps of more than 50 miles.’

9. Page 6, line 10: What does “average annual daily traffic for a road segment” encompass? Personal vehicles? All vehicles?

We have clarified what is included in the average annual daily traffic on Page 56lines 9-12:

“Road segments are weighted by their annual average daily traffic (daily traffic volume averaged over a full year)^[37]. Additionally, we exclude truck-specific annual average daily traffic for the consecutive coverage metric when calculating light-duty vehicle traffic. See Methods for more details.”

10. Page 7, line 1: This sentence should not get cut off like this. Consider changing the order of the equation and figure to prevent this.

We have fixed the formatting issue.

11. Page 9, line 14: “Out-of-state spillover” needs clarification. Discuss this in the Appendix.

We have added the text on page 10 lines 6-8: “We define out-of-state spillover as the percentage of coverage from charging stations on roads in another state but within 500 miles of a given county and consecutively covered.”

12. Page 16, lines 4-7: I appreciate the discussion on this point.

We appreciate the in-depth feedback.

13. Page 17: Consider mapping changes with and without Tesla chargers for clearer understanding.

We map the state of coverage, including and excluding Tesla chargers, in Appendix D on page 33, with the left column of maps showing consecutive charger coverage without Tesla charging stations and the right column of maps showing consecutive charger coverage with Tesla stations.

14. Page 20, Table 2: Add a line between current conditions (2023) and future scenarios.

A line has been added on page 7 in the table.

15. Page 22, Figure 2 caption, and Page 21, lines 22-23: Ensure consistency in describing colors.

The Page 17 line 32 to Page 18 line 2 now reads: “All public non-proprietary charging stations that are level-2 or higher power are mapped in light red and all NEVI-compliant stations with at least four DC fast chargers (41%) are shown in dark blue.”

It is now consistent with the caption.

16. Reference number 2: Ensure the citation includes trips above 500 miles. If using NHTS survey data to calculate, the citation should be: U.S. Department of Transportation, Federal Highway Administration, 2022 National Household Travel Survey. <https://nhts.ornl.gov>

We corrected the citation to be⁵. Found from the U.S. Department of Transportation, Bureau of Transportation Statistics: Distribution of Trips by Distance: National, State, and County level

17. Reference number 7: Provide a URL for the reference.

A URL has been added to the citation³⁶

18. References 31, 34, and 35: Check and fix the URL links.

The DOI link in reference⁴ works. The URL in reference³⁸ has been updated as <https://afdc.energy.gov/stations#/find/nearest>. The URL in reference³⁹ has been updated as https://www.fhwa.dot.gov/environment/alternative_fuel_corridors/deployment_plan/.

19. Page 27, lines 9-11: For the 31 roads that are AFCs but not in the NHS dataset, discuss why it was not possible to take the average annual traffic from connecting segments that are part of the NHS.

We had not previously considered this approach. All analyses and maps have been re-created to take this approach into account. The language in the Appendix at page 36, lines 8-10 has been updated:

“Some roads (31 roads across 11 states) are AFCs, but are not in the NHS dataset; any AFC roads not in the AFC dataset were assigned the average traffic of its connecting road segments and appended to the NHS dataset.” The language on page 16, lines 17-19 has been updated:

“We include the 31 AFC non-NHS roads in the US in this analysis and assign the average traffic of all connected road segments to each AFC.”

Reviewer #3 (Remarks to the Author):

This paper introduces a novel metric named “consecutive coverage” to assess the feasibility of long-distance travel across the U.S. states, which in turn provides an indication to the accessibility of EV charging infrastructure. The work done by the authors is interesting and highlights a new perspective of EV charging infrastructure coverage.

Description of the conducted scenarios were more concrete in later sections after the “Scenarios” section. However, since the “Scenarios” section appears first, I recommend that the authors revise it to indicate the characteristics of each scenario clearly.

We have moved the detailed scenarios description into the scenarios subsection in the introduction for clarity on page 7 lines 10-15:

“We apply our consecutive coverage metric under three scenarios: *Minimum Viable Coverage*, *Fast Charger Coverage*, and *AFCs Reach NEVI-Compliant Status*. Table 1 shows the parameters for each scenario: The *Minimum Viable Coverage* scenario considers all planned and existing non-proprietary public level-2 and DC fast charging stations. The *Fast Charger Coverage* scenario considers all planned and existing NEVI-compliant charging stations. To be NEVI-compliant, stations must have at least four chargers which makes them unlikely to have significant wait times. The *AFCs Reach NEVI-Compliant Status* scenario models coverage of existing and planned NEVI-compliant stations, once they are fully built out along all AFCs. ”

Table 2: Outline of the parameters for the three scenarios modeled in this analysis.

	Minimum Viable Coverage	Fast Charger Coverage	AFCs Reach NEVI-compliant Status
charger accessibility	All public non-proprietary plugs (non-Tesla plugs)		
minimum requirement	any number of L2 & DCFC ports (including <4)	≥4 DCFC ports (NEVI-compliant)	≥4 DCFC ports (NEVI-program complete)
timeline	Current conditions (2023)		Future scenario (NEVI-program complete)
type of charger	L2 & DCFC	DCFC	DCFC
wait-time scenario	If long waits to charge are acceptable	Charging experience closer to re-fueling at gas stations	

I also recommend that the authors discuss in the conclusion section how the “consecutive coverage” metric can be generalized to include other weights rather than the average annual daily traffic. For example: congestion level, temperature, ..., etc. This can be a further progression for this metric to compare between EV charging infrastructure coverage in cities with different geographical context.

We have added a discussion of other weights to Appendix A on future work on Page 26: “Future research or extensions of this work could calculate the consecutive coverage metric, weighting road segments by peak traffic per year or congestion levels. Another extension of this paper could dynamically define the minimum number of chargers per station needed by the annual average daily traffic or peak daily traffic of the year (i.e., requiring more charging ports for high-trafficked roads). Therefore, not only identifying gaps in coverage, but segments of road that contain an insufficient number of chargers per station. As more consumers adopt electric vehicles, queuing times and high-congestion travel days may become a primary form of charging anxiety. Therefore, extending this work to look specifically at peak congestion both for weighting the road segments in the coverage metric and for considering the sufficiency of chargers per station would be a valuable extension of this work.”

Reviewer #4 (Remarks to the Author):

We thank you for your time and participation in the review and reviewer mentorship process.

References

1. Bräun, T. *et al.* Determining the optimal electric vehicle DC-charging infrastructure for Western Australia. *Transportation Research Part D: Transport and Environment* **84**, 102250. ISSN: 1361-9209. doi:<https://doi.org/10.1016/j.trd.2020.102250>. <https://www.sciencedirect.com/science/article/pii/S1361920919308740> (2020).
2. Jochem, P. *et al.* How many fast-charging stations do we need along European highways? *Transportation Research Part D: Transport and Environment* **73**, 120–129. ISSN: 1361-9209. doi:<https://doi.org/10.1016/j.trd.2019.06.005>. <https://www.sciencedirect.com/science/article/pii/S1361920919300215> (2019).
3. Xie, F. *et al.* Long-term strategic planning of inter-city fast charging infrastructure for battery electric vehicles. *Transportation Research Part E: Logistics and Transportation Review* **109**, 261–276. ISSN: 1366-5545. doi:<https://doi.org/10.1016/j.tre.2017.11.014> (2018).
4. Xu, M. *et al.* Mitigate the range anxiety: Siting battery charging stations for electric vehicle drivers. *Transportation Research Part C: Emerging Technologies* **114**, 164–188. ISSN: 0968-090X. doi:<https://doi.org/10.1016/j.trc.2020.02.001> (2020).
5. U.S. Department of Transportation, B. o. T. S. *Distribution of Trips by Distance: National, State, and County level* <https://www.bts.gov/browse-statistical-products-and-data/covid-related/distribution-trips-distance-national-state-and>.
6. Federal Highway Administration. *National Electric Vehicle Infrastructure Formula Program* https://www.fhwa.dot.gov/bipartisan-infrastructure-law/nevi_formula_program.cfm.
7. Hidrue, M. *et al.* Willingness to pay for electric vehicles and their attributes. *Resource and Energy Economics* **33**, 686–705. ISSN: 0928-7655. doi:<https://doi.org/10.1016/j.reseneeco.2011.02.002>. <https://www.sciencedirect.com/science/article/pii/S0928765511000200> (2011).
8. Melaina, M. *et al.* Consumer Convenience and the Availability of Retail Stations as a Market Barrier for Alternative Fuel Vehicles. <https://www.nrel.gov/docs/fy13osti/56898.pdf> (2013).
9. U.S. Department of Energy. *National Alternative Fuels Corridors* <https://afdc.energy.gov/laws/11675>.
10. Descant, S. *Forget Range Anxiety. Most EV Drivers Have Charge Anxiety.* <https://www.govtech.com/fs/forget-range-anxiety-most-ev-drivers-have-charge-anxiety>.
11. Liu, X. *et al.* Comprehensive assessment for different ranges of battery electric vehicles: Is it necessary to develop an ultra-long range battery electric vehicle? *iScience* (2023).
12. Pevec, D. *et al.* A survey-based assessment of how existing and potential electric vehicle owners perceive range anxiety. *Journal of Cleaner Production* **276**, 122779. ISSN: 0959-6526. doi:<https://doi.org/10.1016/j.jclepro.2020.122779>. <https://www.sciencedirect.com/science/article/pii/S0959652620328249> (2020).
13. Melliger, M. A. *et al.* Anxiety vs reality – Sufficiency of battery electric vehicle range in Switzerland and Finland. *Transportation Research Part D: Transport and Environment* **65**, 101–115. ISSN: 1361-9209. doi:<https://doi.org/10.1016/j.trd.2018.08.011>. <https://www.sciencedirect.com/science/article/pii/S1361920917310295> (2018).

14. Agency, I. E. Net Zero Roadmap: A Global Pathway to Keep the 1.5 °C Goal in Reach. *IEA*. <https://www.iea.org/reports/net-zero-roadmap-a-global-pathway-to-keep-the-15-0c-goal-in-reach> (2023).
15. Environmental Protection Agency. *Sources of Greenhouse Gas Emissions* <https://www.epa.gov/ghgemissions/sources-greenhouse-gas-emissions>.
16. Muratori, M. *et al.* The rise of electric vehicles—2020 status and future expectations. *Progress in Energy* **3**, 022002. doi:[10.1088/2516-1083/abe0ad](https://doi.org/10.1088/2516-1083/abe0ad) <https://dx.doi.org/10.1088/2516-1083/abe0ad> (Mar. 2021).
17. Mulrow, J. & Grubert, E. Greenhouse gas emissions embodied in electric vehicle charging infrastructure: a method and case study of Georgia, US 2021–2050. *Environmental Research: Infrastructure and Sustainability* **3**, 015013. doi:[10.1088/2634-4505/acc548](https://doi.org/10.1088/2634-4505/acc548) <https://dx.doi.org/10.1088/2634-4505/acc548> (Mar. 2023).
18. Mulrow, J. *et al.* Sociotechnical convex hulls and the evolution of transportation activity: A method and application to US travel survey data. *Technological Forecasting and Social Change* **149**, 119789. ISSN: 0040-1625. doi:<https://doi.org/10.1016/j.techfore.2019.119789> <https://www.sciencedirect.com/science/article/pii/S0040162519309059> (2019).
19. Zoellick, J. C. & Bisht, A. It’s not (all) about efficiency: Powering and organizing technology from a degrowth perspective. *Journal of Cleaner Production* **197**. Technology and Degrowth, 1787–1799. ISSN: 0959-6526. doi:<https://doi.org/10.1016/j.jclepro.2017.03.234> <https://www.sciencedirect.com/science/article/pii/S095965261730642X> (2018).
20. Aguilar Lopez, F. *et al.* *On the potential of vehicle-to-grid and second-life batteries to provide energy and material security* 2024. doi:<https://doi.org/10.1038/s41467-024-48554-0>.
21. Ren, Y. *et al.* Hidden delays of climate mitigation benefits in the race for electric vehicle deployment. *Nature Communications* **14**. doi:[Ren, Y. , Sun, X. , Wolfram, P. et al. Hiddendelays of climatemitigationbenefitsinther NatCommun14, 3164\(2023\) .https://doi.org/10.1038/s41467-023-38182-5](https://doi.org/10.1038/s41467-023-38182-5) (2023).
22. Lanz, L. *et al.* Comparing the levelized cost of electric vehicle charging options in Europe. *Nature Communications* **13**. doi:<https://doi.org/10.1038/s41467-022-32835-7> (2022).
23. National Renewable Energy Laboratory. *TEMPO: Transportation Energy Mobility Pathway Options Model* <https://www.nrel.gov/transportation/tempo-model.html>.
24. Hoehne, C. *et al.* Exploring decarbonization pathways for USA passenger and freight mobility. *Nature Communications* **14**. doi:<https://doi.org/10.1038/s41467-023-42483-0> (2023).
25. National Renewable Energy Laboratory. *Electric Vehicle Infrastructure Projection Tool (EVI-Pro) Lite* <https://afdc.energy.gov/evi-pro-lite>.
26. National Renewable Energy Laboratory. *EVI-X* <https://www.nrel.gov/transportation/evi-x.html>.
27. Wood, E. *et al.* *The 2030 National Charging Network: Estimating U.S. Light-Duty Demand for Electric Vehicle Charging Infrastructure* <https://www.nrel.gov/docs/fy23osti/85654.pdf>.
28. Ziefle, M. *et al.* Public Perception and Acceptance of Electric Vehicles: Exploring Users’ Perceived Benefits and Drawbacks. doi:https://doi.org/10.1007/978-3-319-07635-5_60 (2014).

29. Higuera-Castillo, E. *et al.* Adoption of electric vehicles: Which factors are really important? *International Journal of Sustainable Transportation* **15**, 799–813. doi:<https://doi.org/10.1080/15568318.2020.1818330>, <https://www.sciencedirect.com/science/article/pii/S1556831822004075> (2021).
30. Vergis, S. & Chen, B. Comparison of plug-in electric vehicle adoption in the United States: A state by state approach. *Research in Transportation Economics* **52**. Sustainable Transportation, 56–64. doi:<https://doi.org/10.1016/j.retrec.2015.10.003>, <https://www.sciencedirect.com/science/article/pii/S0739885915000475> (2015).
31. North American Council for Freight Efficiency. The Case for MD Box Trucks. <https://nacfe.org/research/run-on-less/run-on-less-electric/md-box-trucks/> (2024).
32. Borlaug, B. *et al.* Charging needs for electric semi-trailer trucks. *Renewable and Sustainable Energy Transition* **2**, 100038. ISSN: 2667-095X. doi:<https://doi.org/10.1016/j.rset.2022.100038>, <https://www.sciencedirect.com/science/article/pii/S2667095X22000228> (2022).
33. Bureau, U. C. *Median Income in the Past 12 Months (in 2022 Inflation-Adjusted Dollars)* <https://data.census.gov/table/ACSST1Y2022.S1903?q=income>.
34. The White House. *FACT SHEET: Biden-Harris Administration Announces New Private and Public Sector Investments for Affordable Electric Vehicles* <https://www.whitehouse.gov/briefing-room/statements-releases/2023/04/17/fact-sheet-biden-harris-administration-announces-new-private-and-public-sector-investments-for-affordable-electric-vehicles/>.
35. Egbue, O. & Long, S. Barriers to widespread adoption of electric vehicles: An analysis of consumer attitudes and perceptions. *Energy Policy* **48**. Special Section: Frontiers of Sustainability, 717–729. ISSN: 0301-4215. doi:<https://doi.org/10.1016/j.enpol.2012.06.009>, <https://www.sciencedirect.com/science/article/pii/S0301421512005162> (2012).
36. Greene, D. L. *et al.* *Quantifying the Tangible Value of Public Electric Vehicle Charging Infrastructure* <https://www.energy.ca.gov/publications/2020/quantifying-tangible-value-public-electric-vehicle-charging-infrastructure>.
37. Federal Highway Administration. *National Highway System Shapefile* https://www.fhwa.dot.gov/planning/national_highway_system/nhs_maps/.
38. U.S. Department of Energy. *Alternative Fueling Station Locator* <https://afdc.energy.gov/stations#/find/nearest>.
39. Federal Highway Administration. *Planning, Environment, Realty* https://www.fhwa.dot.gov/environment/alternative_fuel_corridors/deployment_plan/.

REVIEWER COMMENTS

Reviewer #2 (Remarks to the Author):

Thank you for the major improvements to the manuscript. The current version currently addresses all comments we had and improves the results significantly. While we would have like to see more detail discussion about environmental impacts, the paper is well produced and clearly responds to the research gap and questions identified by authors.

The addition in the section on NREL and EVI-Pro was needed, and the authors have tried to tie their work as a more nuanced assessment of long-distance charging station coverage. I agree with the authors on how their metric can be useful in identifying charging station adequacy and for comparing across counties, states, and policies. Their consecutive coverage metric can be especially useful for decision-makers when deciding where to build charging infrastructure. However, the authors did not expand on how this would play out in a policy decision. For instance, if a state has a limited budget for building charging infrastructure, how would they decide between meeting the EVI-Pro metric or the consecutive coverage metric? What is the decision lever that can be used to compare and optimize these two competing metrics? Is it the maximum utility (which would prioritize charging infrastructure in metropolitan areas) or the maximum connectedness (which would prioritize the consecutive coverage metric)? A Pareto comparison between these two metrics (number of charging stations needed in metropolitan areas to meet community-level charging demand and consecutive coverage metrics) is a clear next step in this field. By doing this, decision-makers will have the tools needed to assess their bias toward favoring one or the other, which will eventually result in robust solutions that address multiple needs. While this is likely outside the scope of this particular paper, it is a probable next step that will make this research incredibly useful.

Thank you for the medium and heavy-duty trucks consecutive coverage analysis. It highlights a major gap in EV charging infrastructure. I am surprised the results from this analysis were not highlighted in the main text. This sentence in the appendix is an important finding that I believe needs to be highlighted in the Discussion or other relevant section: "California reaches the highest state-level charging station coverage for both medium and heavy-duty vehicles (24% in both cases), while 37 states have state-level coverage below 10% for medium-duty and 40 states have coverage below 10% for heavy-duty. Charging station access and investment for freight lag significantly behind that of light-duty vehicles."

Thank you for the analysis on the equity implications. While it is harder to dissect the implications, there are some important findings such as: "...leaving the most car-dependent and low household income counties largely with lower coverage overall." Future research might want to look into these equity implications in more detail with other sociodemographic characteristics. The section added in the main text highlights how this consecutive coverage can be achieved in phases by calculating the number of charging stations needed between full coverage of AFCs and full coverage of NHS roads.

Comment 1-6: Thank you for addressing each of these comments.

Comment 7: I think the explanation consecutive coverage metric concerned about sufficiency and not optimization is reasonable, but this limits the applicability of this research in real-world.

Comment 8 & 9: I appreciate the improvements on this explanation.

Comment 13: These maps are clearer compared to the previous iterations.

Comment 19: Since the number of road segments affected by this change is small, I am unable to identify the impact this had on the results, but I appreciate the thoroughness.

Reviewer #3 (Remarks to the Author):

The authors have addressed my concerns, and I have no further comments.

Reviewer #4 (Remarks to the Author):

NCOMMS-23-48883-T

Response to reviewers:

Reviewer #1 (Remarks to the Author):

- Thank you for the major improvements to the manuscript. The current version currently addresses all comments we had and improves the results significantly. While we would have like to see more detail discussion about environmental impacts, the paper is well produced and clearly responds to the research gap and questions identified by authors.

We have added an explanation of further work that could be conducted on the environmental impacts of EV adoption induced by EV infrastructure in the Discussion on page 14, lines 18-20:

“Additionally, models of EV adoption, such as NREL’s TEMPO model, could be used to extend our consecutive coverage metric to find the influence on EV adoption of charging station coverage and the resulting change in lifecycle emissions from higher EV adoption, including changes to air pollutants.”

- The addition in the section on NREL and EVI-Pro was needed, and the authors have tried to tie their work as a more nuanced assessment of long-distance charging station coverage. I agree with the authors on how their metric can be useful in identifying charging station adequacy and for comparing across counties, states, and policies. Their consecutive coverage metric can be especially useful for decision-makers when deciding where to build charging infrastructure. However, the authors did not expand on how this would play out in a policy decision. For instance, if a state has a limited budget for building charging infrastructure, how would they decide between meeting the EVI-Pro metric or the consecutive coverage metric? What is the decision lever that can be used to compare and optimize these two competing metrics? Is it the maximum utility (which would prioritize charging infrastructure in metropolitan areas) or the maximum connectedness (which would prioritize the consecutive coverage metric)? A Pareto comparison between these two metrics (number of charging stations needed in metropolitan areas to meet community-level charging demand and consecutive coverage metrics) is a clear next step in this field. By doing this, decision-makers will have the tools needed to assess their bias toward favoring one or the other, which will eventually result in robust solutions that address multiple needs. While this is likely outside the scope of this particular paper, it is a probable next step that will make this research incredibly useful.

We would like to clarify that EVI-Pro is not an optimization; it is an estimate of the number of charging stations needed and the associated electricity load of a given metropolitan statistical area. Our consecutive coverage metric highlights gaps in EV charging infrastructure coverage but, similarly, does not optimize EV charger siting. A Pareto frontier is a useful tool when optimizing with two objective functions to find the trade-offs between the two objectives (e.g., with a limited budget, the trade-offs on EV adoption influence between investing in community-level and long-distance charging stations). Due to neither model using optimization, a Pareto frontier is outside the scope of this paper. Although a Pareto frontier is outside of the scope of this work, we mention the potential in our Future work section on the Supplementary Information page 1, lines 31-34:

“Future work could look at the trade-offs in EV adoption impacts between investing in community-level charging station access vs. long-distance charging station coverage. An optimization of community-level and long-distance charging needs could yield a Pareto frontier of ideal charging station distribution between local access and long-distance coverage.”

- Thank you for the medium and heavy-duty trucks consecutive coverage analysis. It highlights a major gap in EV charging infrastructure. I am surprised the results from this analysis were not highlighted in the main text. This sentence in the appendix is an important finding that I believe needs to be highlighted in the Discussion or other relevant section: “California reaches the highest state-level charging station coverage for both medium and heavy-duty vehicles (24% in both cases), while 37

states have state-level coverage below 10% for medium-duty and 40 states have coverage below 10% for heavy-duty. Charging station access and investment for freight lag significantly behind that of light-duty vehicles.

We have added some medium and heavy-duty findings to our discussion on page 14, lines 21-23:

“Charging station access for freight lags behind that of light-duty vehicles, despite heavy-duty trucks compromising a significant portion of interstate traffic¹¹. California reaches the highest state-level charging station coverage for both medium- and heavy-duty coverage (24%), while 40 states have heavy-duty coverage below 10% (Supplementary Figs. 1-3).”

- Thank you for the analysis on the equity implications. While it is harder to dissect the implications, there are some important findings such as: “...leaving the most car-dependent and low household income counties largely with lower coverage overall.” Future research might want to look into these equity implications in more detail with other sociodemographic characteristics. The section added in the main text highlights how this consecutive coverage can be achieved in phases by calculating the number of charging stations needed between full coverage of AFCs and full coverage of NHS roads.

We have added the need for further research on the equity implications of long-distance charging station coverage to our discussion on Page 15, lines 1-3:

“Fast charging coverage increases with median household income at the county level (Supplementary Fig. 6). Future work should investigate the equity implications of long-distance charging station access on EV adoption accessibility.”

- Comment 1-6: Thank you for addressing each of these comments.
Comment 7: I think the explanation consecutive coverage metric concerned about sufficiency and not optimization is reasonable, but this limits the applicability of this research in real-world.
Comment 8 & 9: I appreciate the improvements on this explanation.
Comment 13: These maps are clearer compared to the previous iterations. Comment 19: Since the number of road segments affected by this change is small, I am unable to identify the impact this had on the results, but I appreciate the thoroughness.

Thank you for your time and feedback.

Reviewer #3 (Remarks to the Author):

The authors have addressed my concerns, and I have no further comments.

Thank you for your time in reviewing this paper.

Reviewer 4 (Remarks to the Author):

Thank you for your time in reviewing this paper.